# Direct observation of the dead-cone effect in quantum chromodynamics

ALICE Collaboration*✉

In particle collider experiments, elementary particle interactions with large momentum transfer produce quarks and gluons (known as partons) whose evolution is governed by the strong force, as described by the theory of quantum chromodynamics (QCD)[1]. These partons subsequently emit further partons in a process that can be described as a parton shower[2], which culminates in the formation of detectable hadrons. Studying the pattern of the parton shower is one of the key experimental tools for testing QCD. This pattern is expected to depend on the mass of the initiating parton, through a phenomenon known as the dead-cone effect, which predicts a suppression of the gluon spectrum emitted by a heavy quark of mass $m_Q$ and energy $E$, within a cone of angular size $m_Q/E$ around the emitter[3]. Previously, a direct observation of the dead-cone effect in QCD had not been possible, owing to the challenge of reconstructing the cascading quarks and gluons from the experimentally accessible hadrons. We report the direct observation of the QCD dead cone by using new iterative declustering techniques[4,5] to reconstruct the parton shower of charm quarks. This result confirms a fundamental feature of QCD. Furthermore, the measurement of a dead-cone angle constitutes a direct experimental observation of the non-zero mass of the charm quark, which is a fundamental constant in the standard model of particle physics.

In particle colliders, quarks and gluons are produced in high-energy interactions through processes with large momentum transfer, which are calculable and well described by quantum chromodynamics (QCD). These partons undergo subsequent emissions, resulting in the production of more quarks and gluons. This evolution can be described in the collinear limit by a cascade process known as a parton shower, which transfers the original parton energy to multiple lower energy particles. This shower then evolves into a multi-particle final state, with the partons combining into a spray of experimentally detectable hadrons known as a jet[6]. The pattern of the parton shower is expected to depend on the mass of the emitting parton, through a phenomenon known as the dead-cone effect, whereby the radiation from an emitter of mass $m$ and energy $E$ is suppressed at angular scales smaller than $m/E$, relative to the direction of the emitter. The dead-cone effect is a fundamental feature of all gauge field theories (see ref. [3] for the derivation of the dead cone in QCD).

The dead-cone effect is expected to have sizeable implications for charm and beauty quarks, which have masses of $1.28 \pm 0.02$ GeV/$c^2$ and $4.18^{+0.03}_{-0.02}$ GeV/$c^2$ (ref. [1]) in the minimal subtraction scheme, respectively, at energies on the GeV scale. The emission probability in the collinear region, which is the divergent limit of QCD at which the radiation is most intense, is suppressed with increasing mass of the quark. This leads to a decrease in the mean number of particles produced in the parton shower. The DELPHI Collaboration at the LEP $e^+e^-$ collider measured the multiplicity difference between events containing jets initiated by heavy beauty quarks and those containing light quarks (up, down or strange). They found that the differences depend only on the quark mass[7], which was attributed to the suppression of collinear gluon radiation from the heavy quark because of the dead-cone effect. A measurement of the momentum density of jet constituents as a function of distance from the jet axis was also performed by the ATLAS collaboration at CERN[8], which pointed to a depletion of momentum close to the jet axis that was ascribed as a consequence of the dead-cone effect. The mass of the beauty quark was also estimated through a phenomenological fit to the measured data[9]. As hard (large transverse momentum) emissions are preferentially emitted at small angles, and are therefore suppressed for massive emitters, heavy quarks also retain a larger fraction of their original momentum compared to lighter quarks, leading to a phenomenon known as the leading-particle effect. This has been well established experimentally, with the fraction of the jet momentum carried by the leading (highest transverse momentum) hadron containing a charm or beauty quark (heavy-flavour hadron) in jets, peaking at 0.6–0.7 and 0.8–0.9, respectively, whereas the corresponding fraction carried by the leading hadron in light quark-initiated jets peaks at smaller values[10–14].

Until now, a direct experimental measurement of the dead-cone effect has been subject to two main challenges. First, the dead-cone angular region can receive contributions from hadronization effects or particles that do not originate from the gluon radiation from the heavy-flavour quark, such as the decay products of heavy-flavour hadrons. The second difficulty lies in the accurate determination of the dynamically evolving direction of the heavy-flavour quark, relative to which the radiation is suppressed, throughout the shower process. The development of new experimental declustering

techniques[4] enables these aforementioned difficulties to be overcome by reconstructing the evolution of the jet shower, giving access to the kinematic properties of each individual emission. These techniques reorganize the particle constituents of an experimentally reconstructed jet, to access the building blocks of the shower and trace back the cascade process. Isolated elements of the reconstructed parton shower that are likely to be unmodified by hadronization processes provide a good proxy for real quark and gluon emissions (splittings). These reclustering techniques have been demonstrated in inclusive (without tagging the initiating parton flavour) jets to successfully reconstruct splittings that are connected to or that preserve the memory of the parton branchings. This is demonstrated by measurements such as the groomed momentum balance[15–18], which probes the Dokshitzer–Gribov–Lipatov–Altarelli–Parisi splitting function[19], and the Lund plane[20], which exposes the running of the strong coupling with the scale of the splittings. An experimental method to expose the dead cone in boosted top-quark events was also proposed in ref. [21].

Reclustering techniques are extended in this work to jets containing a charm quark based on the prescription given in ref. [22]. These jets are tagged through the presence of a reconstructed $D^0$ meson amongst their constituents, which has a mass of $1.86\ \mathrm{GeV}/c^2$ (ref. [1]) and is composed of a heavy charm quark and a light anti-up quark. The measurement is performed in proton–proton collisions at a centre-of-mass energy of $\sqrt{s} = 13\ \mathrm{TeV}$ at the Large Hadron Collider (LHC), using the ALICE (A Large Ion Collider Experiment) detector. Further details of the detector apparatus and data measured can be found in the Methods. As the charm-quark flavour is conserved through the shower process, this provides an opportunity to isolate and trace back the emission history of the charm quark. In this way, by comparing the emission patterns of charm quarks to those of light quarks and gluons, the QCD dead cone can be directly revealed.

## Selecting jets containing a $D^0$ meson

To select jets initiated by a charm quark, through the presence of a $D^0$ meson in their list of constituents, the $D^0$ mesons and jets need to be reconstructed in the events. The $D^0$-meson candidates (and their antiparticles) were reconstructed in the transverse-momentum interval $2 < p_\mathrm{T}^{D^0} < 36\ \mathrm{GeV}/c$, through the $D^0 \to K^- \pi^+$ (and charged conjugate) hadronic decay channel, which has a branching ratio of $3.95 \pm 0.03\%$ (ref. [1]). The $D^0$-meson candidates were identified by topological selections based on the displacement of the $D^0$-meson candidate decay vertex, in addition to applying particle identification on the $D^0$-meson candidate decay particles. These selection criteria largely suppress the combinatorial background of $K^\mp \pi^\pm$ pairs that do not originate from the decay of a $D^0$ meson. Further details on the selection criteria are provided in ref. [23].

Tracks (reconstructed charged-particle trajectories) corresponding to the $D^0$-meson candidate decay particles were replaced by the reconstructed $D^0$-meson candidate in the event, with the $D^0$-meson candidate four-momentum being the sum of the decay-particle four-momenta. One benefit of this procedure is to avoid the case in which the decay products of the $D^0$-meson candidate fill the dead-cone region. A jet-finding algorithm was then used to cluster the particles (tracks and the $D^0$-meson candidate) in the event, to reconstruct the parton shower by sequentially recombining the shower particles into a single object (the jet). The jet containing the $D^0$-meson candidate was then selected. The four-momentum of the jet is a proxy for the four-momentum of the charm quark initiating the parton shower. The jet-finding algorithm used was the anti-$k_\mathrm{T}$ algorithm[24] from the Fastjet package[25], which is a standard choice for jet reconstruction because of its high performance in reconstructing the original parton kinematics. More details on the jet finding procedure can be found in the Methods.

## Reconstructing the jet shower

Once jets containing a $D^0$-meson candidate amongst their constituents are selected, the internal cascade process is reconstructed. This is done by reorganizing (reclustering) the jet constituents according to the Cambridge–Aachen (C/A) algorithm[26], which clusters these constituents based solely on their angular distance from one another. A pictorial representation of this reclustering process, which starts by reconstructing the smallest angle splittings, is shown in the top panels of Fig. 1. As QCD emissions approximately follow an angular-ordered structure[27], the C/A algorithm was chosen as it also returns an angular-ordered splitting tree.

This splitting tree is then iteratively declustered by unwinding the reclustering history, to access the building blocks of the reconstructed jet shower. At each declustering step, two prongs corresponding to a splitting are returned. The angle between these splitting daughter prongs, $\theta$, the relative transverse momentum of the splitting, $k_\mathrm{T}$, and the sum of the energy of the two prongs, $E_\mathrm{Radiator}$, are registered. As the charm flavour is conserved throughout the showering process, the full reconstruction of the $D^0$-meson candidate enables the isolation of the emissions of the charm quark in the parton shower, by following the daughter prong containing the fully reconstructed $D^0$-meson candidate at each declustering step. This can be seen in the bottom part of Fig. 1, which shows the evolution of the charm quark reconstructed from the measured final state particles. Moreover, the kinematic properties of the charm quark are updated along the splitting tree, enabling an accurate reconstruction of each emission angle against the dynamically evolving charm-quark direction. It was verified that in more than 99% of the cases the prong containing the $D^0$-meson candidate at each splitting coincided with the leading prong. This means that following the $D^0$-meson candidate or leading prong at each step is equivalent, and therefore a complementary measurement for an inclusive jet sample, when no flavour tagging is available, can be made by following the leading prong through the reclustering history. As the inclusive sample is dominated by massless gluon and nearly massless light quark-initiated jets, it acts as a reference to highlight the mass effects present in the charm tagged sample.

## Extracting the true charm splittings

The selected sample of splittings has contributions from jets tagged with combinatorial $K^\mp \pi^\pm$ pairs, which are not rejected by the applied topological and particle identification selections. The measured invariant mass of real $D^0$ mesons, which corresponds to the rest mass, is distributed in a Gaussian (because of uncertainties in the measurement of the momenta of the $K^\mp \pi^\pm$ pairs) with a peak at the true $D^0$-meson mass. This enables the implementation of a statistical two-dimensional side-band subtraction procedure, which characterizes the background distribution of splittings by sampling the background-dominated regions of the $D^0$-meson candidate invariant mass distributions, far away from the signal peak. In this way the combinatorial contribution can be accounted for and removed. Furthermore, the selections on the $D^0$-meson candidates also select a fraction of $D^0$ mesons originating as a product of beauty-hadron decays. These were found to contribute 10–15% of the reconstructed splittings, with only a small influence on the results, which will be discussed later. The studies were performed using Monte Carlo (MC) PYTHIA 6.425 (Perugia 2011)[28,29] simulations (this generator includes mass effects in the parton shower[30] and was used for all MC-based corrections in this work), propagating the generated particles through a detailed description of the ALICE detector with GEANT3 (ref. [31]). The finite efficiency of selecting real $D^0$-meson tagged jets, through the chosen selection criteria on the $D^0$-meson candidates, as well as kinematic selections on the jets, was studied and accounted for through MC simulations. This efficiency was found to be strongly $p_\mathrm{T}^{D^0}$ dependent and different for $D^0$ mesons originating

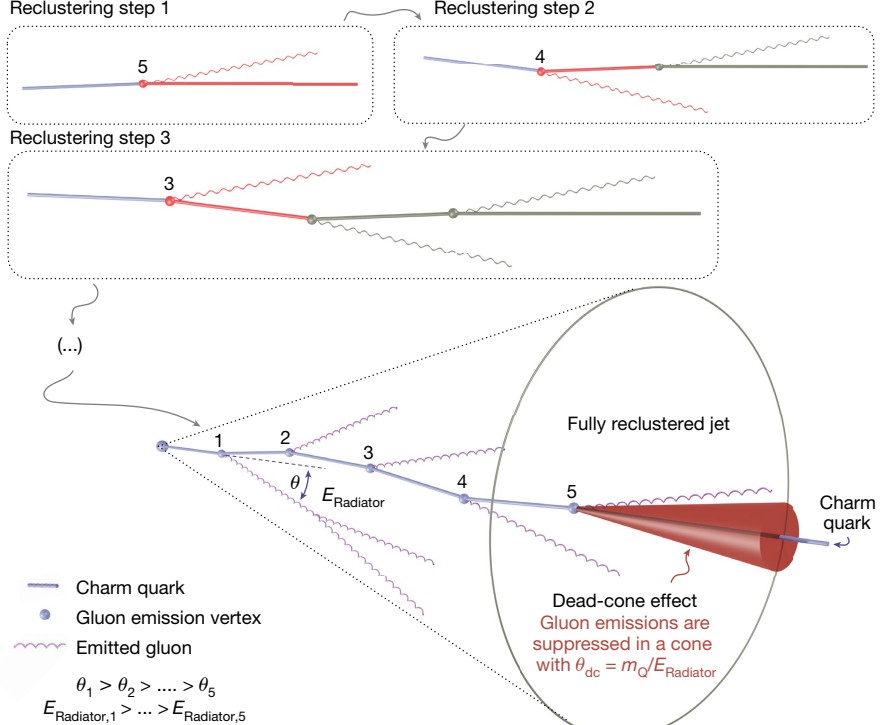

**Fig. 1 | Reconstruction of the showering quark.** A sketch detailing the reconstruction of the showering charm quark, using iterative declustering, is presented. The top panels show the initial reclustering procedure with the C/A algorithm, in which the particles separated by the smallest angles are brought together first. Once the reclustering is complete, the declustering procedure is carried out by unwinding the reclustering history. Each splitting node is numbered according to the declustering step in which it is reconstructed. With each splitting, the charm-quark energy, $E_{\text{Radiator},n}$, is reduced and the gluon is emitted at a smaller angle, $\theta_n$, with respect to previous emissions. The mass of the heavy quark, $m_Q$, remains constant throughout the showering process. At each splitting, gluon emissions are suppressed in the dead-cone region (shown by a red cone for the last splitting), which increases in angle as the quark energy decreases throughout the shower.

from the hadronization of charm quarks or from the decay of beauty hadrons. Further details on these analysis steps can be found in the Methods.

As the reconstructed jet shower is built from experimentally detectable hadrons, as opposed to partons, hadronization effects must be accounted for. As hadronization processes occur at low non-perturbative scales, they are expected to distort the parton shower by mainly adding low-$k_T$ splittings[32]. A selection of $k_T > 200$ MeV/$c$ ensures that only sufficiently hard splittings are accepted and is used to suppress such hadronization effects. Other choices of $k_T$ selection were also explored, with stronger $k_T$ selections further removing non-perturbative effects from the measurement, at the expense of statistical precision. Other non-perturbative effects, such as the underlying event, contribute with extra soft splittings primarily at large angles and do not affect the small-angle region under study.

Detector effects also distort the reconstructed parton shower through inefficiencies and irresolution in the tracking of charged particles. However, these have been tested and largely cancel in the final observable, and any residual effects are quantified in a data-driven way and included in the systematic uncertainties.

It should be noted that in addition to direct heavy-flavour pair creation in the elementary hard scattering, charm quarks can also be produced in higher-order processes as a result of gluon splitting. Therefore, the shower history of $D^0$ mesons containing such charm quarks will also have contributions from splittings originating from gluons. Furthermore, in the case of high transverse momentum gluons in which the charm quarks are produced close in angle to each other, the dead-cone region of the charm quark hadronizing into the reconstructed $D^0$ meson can be populated by particles produced in the shower, hadronization and subsequent decays of the other (anti-)charm quark. The influence of such contaminations through gluon splittings was studied with MC simulations and found to be negligible.

## The observable $R(\theta)$

The observable used to reveal the dead cone is built by constructing the ratio of the splitting angle ($\theta$) distributions for $D^0$-meson tagged jets and inclusive jets, in bins of $E_{\text{Radiator}}$. This is given by

$$R(\theta) = \frac{1}{N^{D^0\,\text{jets}}} \frac{dn^{D^0\,\text{jets}}}{d\ln(1/\theta)} \bigg/ \frac{1}{N^{\text{inclusive jets}}} \frac{dn^{\text{inclusive jets}}}{d\ln(1/\theta)} \bigg|_{k_T, E_{\text{Radiator}}} \tag{1}$$

where the $\theta$ distributions were normalized to the number of jets that contain at least one splitting in the given $E_{\text{Radiator}}$ and $k_T$ selection, denoted by $N^{D^0\,\text{jets}}$ and $N^{\text{inclusive jets}}$ for the $D^0$-meson tagged and inclusive jet samples, respectively. Expressing equation (1) in terms of the logarithm of the inverse of the angle is natural, given that at leading order the QCD probability for a parton to split is proportional to $\ln(1/\theta)\ln(k_T)$.

A selection on the transverse momentum of the leading track in the leading prong of each registered splitting in the inclusive jet sample, $p_{\text{T,inclusive jets}}^{\text{ch,leading track}} \geq 2.8$ GeV/$c$, was applied. This corresponds to the transverse mass (obtained through the quadrature sum of the rest mass and transverse momentum) of a 2 GeV/$c$ $D^0$ meson and accounts for the $p_T^{D^0}$ selection in the $D^0$-meson tagged jet sample, enabling a fair comparison of the two samples.

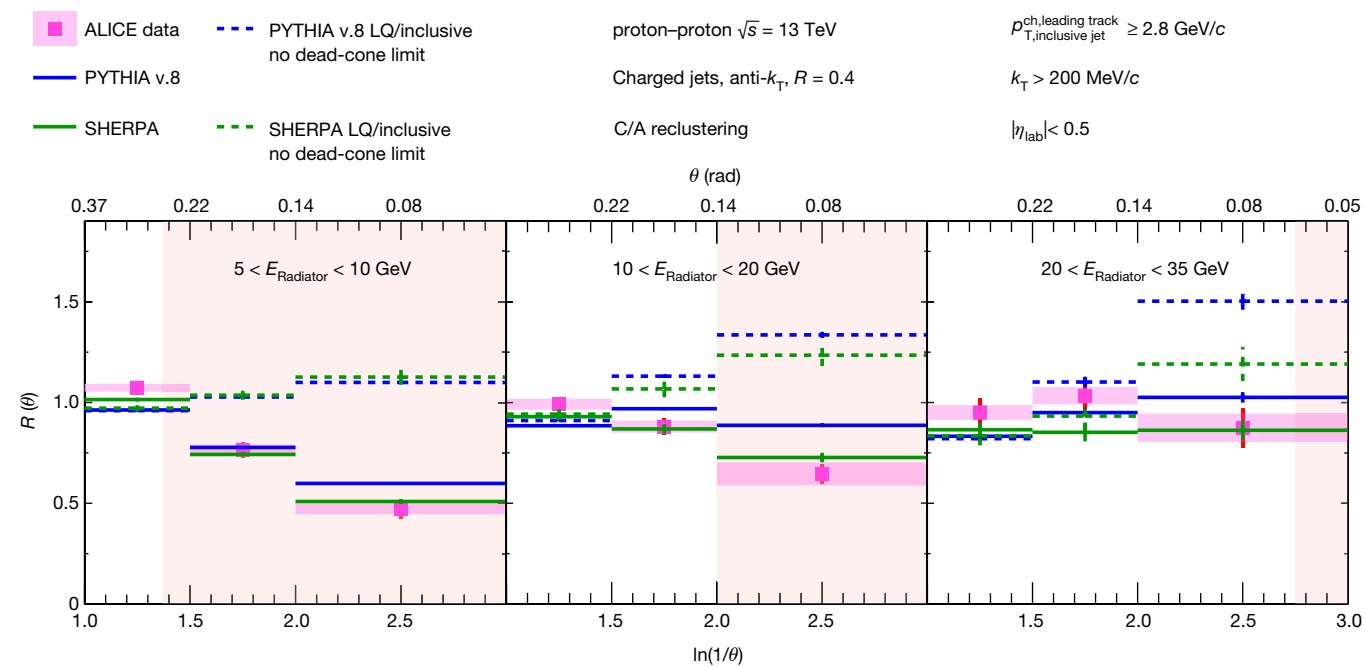

**Fig. 2 | Ratios of splitting angle probability distributions.** The ratios of the splitting-angle probability distributions for $D^0$-meson tagged jets to inclusive jets, $R(\theta)$, measured in proton–proton collisions at $\sqrt{s} = 13$ TeV, are shown for $5 < E_{\text{Radiator}} < 10$ GeV (left panel), $10 < E_{\text{Radiator}} < 20$ GeV (middle panel) and $20 < E_{\text{Radiator}} < 35$ GeV (right panel). The data are compared with PYTHIA v.8 and SHERPA simulations, including the no dead-cone limit given by the ratio of the angular distributions for light-quark jets (LQ) to inclusive jets. The pink shaded areas correspond to the angles within which emissions are suppressed by the dead-cone effect, assuming a charm-quark mass of 1.275 GeV/$c^2$.

In the absence of mass effects, the charm quark is expected to have the same radiating properties as a light quark. In this limit, equation (1) can be rewritten as

$$R(\theta)_{\text{no dead-cone limit}}$$
$$= \frac{1}{N^{\text{LQ jets}}} \frac{dn^{\text{LQ jets}}}{d\ln(1/\theta)} \bigg/ \frac{1}{N^{\text{inclusive jets}}} \frac{dn^{\text{inclusive jets}}}{d\ln(1/\theta)} \bigg|_{k_T, E_{\text{Radiator}}}, \quad (2)$$

where the superscript LQ refers to light quarks, and the inclusive sample contains both light-quark and gluon-initiated jets. This indicates that the $R(\theta)_{\text{no dead-cone limit}}$ ratio depends on the differences between light-quark and gluon radiation patterns, which originate from the fact that gluons carry two colour charges (the charge responsible for strong interactions) whereas quarks only carry one. These differences result in quarks fragmenting at a lower rate and more collinearly than gluons. Therefore, in the limit of having no dead-cone effect, the ratio of the $\theta$ distributions for $D^0$-meson tagged jets and inclusive jets becomes $R(\theta)_{\text{no dead-cone limit}} > 1$, at small angles. This was verified through SHERPA v.2.2.8 (ref. [33]) and PYTHIA v.8.230 (Tune 4C)[34] MC generator calculations, with the specific $R(\theta)_{\text{no dead-cone limit}}$ value dependent on the quark and gluon fractions in the inclusive sample. SHERPA and PYTHIA are two MC generators commonly used in high-energy particle physics and they use different shower prescriptions and hadronization models. Both models implement the dead-cone effect.

## Exposing the dead cone

The measurements of $R(\theta)$, in the three radiator (charm-quark) energy intervals $5 < E_{\text{Radiator}} < 10$ GeV, $10 < E_{\text{Radiator}} < 20$ GeV and $20 < E_{\text{Radiator}} < 35$ GeV, are presented in Fig. 2. Detector effects largely cancel out in the ratio and results are compared to particle-level simulations. Residual detector effects are considered in the systematic uncertainty together with uncertainties associated with the reconstruction and signal extraction of $D^0$-meson tagged jets, as well as detector inefficiencies in the reconstruction of charged tracks in both the $D^0$-meson tagged and inclusive jet samples. More details on the study of systematic uncertainties can be found in the Methods.

A significant suppression in the rate of small-angle splittings is observed in $D^0$-meson tagged jets relative to the inclusive jet population. In Fig. 2, the data are compared with particle-level SHERPA (green) and PYTHIA v.8.230 (blue) MC calculations, with SHERPA v.2.2.8 providing a better agreement with the data. The no dead-cone baseline, as described in equation (2), is also provided for each MC generator (dashed lines). The suppression of the measured data points relative to the no dead-cone limit directly reveals the dead cone within which the charm-quark emissions are suppressed. The coloured regions in the plots correspond to the dead-cone angles in each $E_{\text{Radiator}}$ interval, $\theta_{\text{dc}} < m_Q / E_{\text{Radiator}}$, where emissions are suppressed. For a charm-quark mass $m_Q = 1.275$ GeV/$c^2$ (ref. [1]), these angles correspond to $\ln(1/\theta_{\text{dc}}) \gtrsim 1.37$, 2 and 2.75 for the intervals $5 < E_{\text{Radiator}} < 10$ GeV, $10 < E_{\text{Radiator}} < 20$ GeV and $20 < E_{\text{Radiator}} < 35$ GeV, respectively. These values are in qualitative agreement with the angles at which the data start to show suppression relative to the MC limits for no dead-cone effect. The magnitude of this suppression increases with decreasing radiator energy, as expected from the inverse dependence of the dead-cone angle on the energy of the radiator.

A lower limit for the significance of the small-angle suppression is estimated by comparing the measured data to $R(\theta) = 1$, which represents the limit of no dead-cone effect in the case in which the inclusive sample is entirely composed of light quark-initiated jets. To test the compatibility of the measured data with the $R(\theta) = 1$ limit, a statistical test was performed by generating pseudodata distributions consistent with the statistical and systematic uncertainties of the measured data. A chi-square test was then carried out against this hypothesis for each of the pseudodata distributions. The mean $P$ values correspond to significances of $7.7\sigma$, $3.5\sigma$ and $1.0\sigma$, for the $5 < E_{\text{Radiator}} < 10$ GeV, $10 < E_{\text{Radiator}} < 20$ GeV and $20 < E_{\text{Radiator}} < 35$ GeV intervals, respectively. A $\sigma$ value greater than 5 is considered the criteria for a definitive observation, whereas the value of 1.0 is consistent with the null hypothesis.

The MC distributions shown were generated separately for prompt (charm-quark initiated) and non-prompt (beauty-quark initiated) $D^0$-meson tagged jet production and were then combined using the prompt and non-prompt fractions in data calculated with POWHEG[35] + PYTHIA v.6.425[34] simulations. The non-prompt fraction was found to be independent of the splitting angle and corresponds to approximately 10% of the splittings in the $5 < E_{Radiator} < 10$ GeV interval and approximately 15% of the splittings in both the $10 < E_{Radiator} < 20$ GeV and $20 < E_{Radiator} < 35$ GeV intervals. It was verified through the MC simulations that non-prompt $D^0$-meson tagged jets should exhibit a smaller suppression at small angles in $R(\theta)$ compared with inclusive jets than their prompt counterparts. This is due to the additional decay products accompanying non-prompt $D^0$-meson tagged jets that are produced in the decay of the beauty hadron. These may populate the dead-cone region, leading to a smaller observed suppression in $R(\theta)$, despite the larger dead-cone angle of the heavier beauty quark.

## Conclusions

We have reported the direct measurement of the QCD dead cone, using iterative declustering of jets tagged with a fully reconstructed charmed hadron. The dead cone is a fundamental phenomenon in QCD, dictated by the non-zero quark masses, whose direct experimental observation has previously remained elusive. This measurement provides insight into the influence of mass effects on jet properties and provides constraints for MC models. These results pave the way for a study of the mass dependence of the dead-cone effect, by measuring the dead cone of beauty jets tagged with a reconstructed beauty hadron.

A future study of the dead-cone effect in heavy-ion collisions, in which partons interact strongly with the hot QCD medium that is formed and undergo energy loss through (dominantly) medium-induced radiation, is also envisaged. If a dead cone were observed for these medium-induced emissions, it would be a confirmation of the theoretical understanding of in-medium QCD radiation, which is a primary tool used to characterize the high-temperature phase of QCD matter[36–38].

The quark masses are fundamental constants of the standard model of particle physics and needed for all numerical calculations within its framework. Because of confinement, their values are commonly inferred through their influence on hadronic observables. An exception is the top quark, which decays before it can hadronize, as its mass can be constrained experimentally from the direct reconstruction of the decay final states[39] (see ref. [40] for a review of top mass measurements at the Fermilab Tevatron and CERN LHC).

By accessing the kinematics of the showering charm quark, before hadronization, and directly uncovering the QCD dead-cone effect, our measurement provides direct sensitivity to the mass of quasi-free charm quarks, before they bind into hadrons.

Furthermore, future high-precision measurements using this technique on charm and beauty tagged jets, potentially in conjunction with machine-learning tools to separate quark and gluon emissions, could experimentally constrain the magnitude of the quark masses.

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

## ALICE Collaboration

S. Acharya[1], D. Adamova[2], A. Adler[3], J. Adolfsson[4], G. Aglieri Rinella[5], M. Agnello[6], N. Agrawal[7], Z. Ahammed[1], S. Ahmad[8], S. U. Ahn[9], I. Ahuja[10], Z. Akbar[11], A. Akindinov[12], M. Al-Turany[13], S. N. Alam[14], D. Aleksandrov[15], B. Alessandro[16], H. M. Alfanda[17], R. Alfaro Molina[18], B. Ali[8], Y. Ali[19], A. Alici[20], N. Alizadehvandchali[21], A. Alkin[5], J. Alme[22], T. Alt[23], L. Altenkamper[22], I. Altsybeev[24], M. N. Anaam[17], C. Andrei[25], D. Andreou[26], A. Andronic[27], M. Angeletti[5], V. Anguelov[28], F. Antinori[29], P. Antonioli[7], C. Anuj[8], N. Apadula[30], L. Aphecetche[31], H. Appelshauser[23], S. Arcelli[20], R. Arnaldi[16], I. C. Arsene[32], M. Arslandok[28,33], A. Augustinus[5], R. Averbeck[13], S. Aziz[34], M. D. Azmi[8], A. Badala[35], Y. W. Baek[36], X. Bai[13,37], R. Bailhache[23], Y. Bailung[38], R. Bala[39], A. Balbino[6], A. Baldisseri[40], B. Balis[41], M. Ball[42], D. Banerjee[43], R. Barbera[44], L. Barioglio[45,46], M. Barlow[47], G. G. Barnafoldi[48], L. S. Barnby[49], V. Barret[50], C. Bartels[51], K. Barth[5], E. Bartsch[23], F. Baruffaldi[52], N. Bastid[50], S. Basu[4], G. Batigne[31], B. Batyunya[53], D. Bauri[54], J. L. Bazo Alba[55], I. G. Bearden[56], C. Beattie[33], I. Belikov[57], A. D. C. Bell Hechavarria[27], F. Bellini[5,20], R. Bellwied[21], S. Belokurova[24], V. Belyaev[58], G. Bencedi[59], S. Beole[46], A. Bercuci[25], Y. Berdnikov[60], A. Berdnikova[28], L. Bergmann[28], M. G. Besoiu[61], L. Betev[5], P. P. Bhaduri[1], A. Bhasin[39], M. A. Bhat[43], B. Bhattacharjee[62], P. Bhattacharya[63], L. Bianchi[46], N. Bianchi[64], J. Bielcik[65], J. Bielcikova[2], J. Biernat[66], A. Bilandzic[45], G. Biro[48], S. Biswas[43], J. T. Blair[67], D. Blau[15], M. B. Blidaru[13], C. Blume[23], G. Boca[68,69], F. Bock[70], A. Bogdanov[58], S. Boi[63], J. Bok[71], L. Boldizsar[48], A. Bolozdynya[58], M. Bombara[10], P. M. Bond[5], G. Bonomi[69,72], H. Borel[40], A. Borissov[73], H. Bossi[33], E. Botta[46], L. Bratrud[23], P. Braun-Munzinger[13], M. Bregant[74], M. Broz[65], G. E. Bruno[75,76], M. D. Buckland[51], D. Budnikov[77], H. Buesching[23], S. Bufalino[6], O. Bugnon[31], P. Buhler[78], Z. Buthelezi[79,80], J. B. Butt[19], S. A. Bysiak[66], D. Caffarri[26], M. Cai[52,17], H. Caines[33], A. Caliva[13], E. Calvo Villar[55], J. M. M. Camacho[81], R. S. Camacho[82], P. Camerini[83], F. D. M. Canedo[74], F. Carnesecchi[5,20], R. Caron[40], J. Castillo Castellanos[40], E. A. R. Casula[63], F. Catalano[6], C. Ceballos Sanchez[53], P. Chakraborty[54], S. Chandra[1], S. Chapeland[5], M. Chartier[51], S. Chattopadhyay[1], S. Chattopadhyay[84], A. Chauvin[63], T. G. Chavez[82], C. Cheshkov[85], B. Cheynis[85], V. Chibante Barroso[5], D. D. Chinellato[86], S. Cho[71], P. Chochula[5], P. Christakoglou[26], C. H. Christensen[56], P. Christiansen[4], T. Chujo[87], C. Cicalo[20], L. Cifarelli[20], F. Cindolo[7], M. R. Ciupek[13], G. Clai[7,150], J. Cleymans[89,154], F. Colamaria[90], J. S. Colburn[49], D. Colella[48,75,76,90], A. Collu[30], M. Colocci[5,20], M. Concas[16,151], G. Conesa Balbastre[92], Z. Conesa del Valle[34], G. Contin[83], J. G. Contreras[65], M. L. Coquet[40], T. M. Cormier[70], P. Cortese[93], M. R. Cosentino[94], F. Costa[5], S. Costanza[68,69], P. Crochet[50], R. Cruz-Torres[30], E. Cuautle[59], P. Cui[17], L. Cunqueiro[70], A. Dainese[29], F. P. A. Damas[31,40], M. C. Danisch[28], A. Danu[61], I. Das[84], P. Das[95], P. Das[43], S. Das[43], S. Dash[54], S. De[95], A. De Caro[96], G. de Cataldo[90], L. De Cilladi[46], J. de Cuveland[97], A. De Falco[63], D. De Gruttola[96], N. De Marco[16], C. De Martin[83], S. De Pasquale[96], S. Deb[38], H. F. Degenhardt[74], K. R. Deja[98], L. Dello Stritto[96], S. Delsanto[46], W. Deng[17], P. Dhankher[99], D. Di Bari[76], A. Di Mauro[5], R. A. Diaz[100], T. Dietel[89], Y. Ding[85,17], R. Divia[5], D. U. Dixit[99], O. Djuvsland[22], U. Dmitrieva[101], J. Do[71], A. Dobrin[61], B. Donigus[23], O. Dordic[32], A. K. Dubey[1], A. Dubla[13,26], S. Dudi[102], M. Dukhishyam[95], P. Dupieux[50], N. Dzalaiova[103], T. M. Eder[27], R. J. Ehlers[70], V. N. Eikeland[22], F. Eisenhut[23], D. Elia[90], B. Erazmus[31], F. Ercolessi[20], F. Erhardt[104], A. Erokhin[24], M. R. Ersdal[22], B. Espagnon[34], G. Eulisse[5], D. Evans[49], S. Evdokimov[105], L. Fabbietti[45], M. Faggin[52], J. Faivre[92], F. Fan[17], A. Fantoni[64], M. Fasel[70], P. Fecchio[6], A. Feliciello[16], G. Feofilov[24], A. Fernandez Tellez[82], A. Ferrero[40], A. Ferretti[46], V. J. G. Feuillard[28], J. Figiel[66], S. Filchagin[77], D. Finogeev[101], F. M. Fionda[22,88], G. Fiorenza[5,75], F. Flor[21], A. N. Flores[67], S. Foertsch[79], P. Foka[13], S. Fokin[15], E. Fragiacomo[91], E. Frajna[48], U. Fuchs[5], N. Funicello[96], C. Furget[92], A. Furs[101], J. J. Gaardhoje[56], M. Gagliardi[46], A. M. Gago[55], A. Gal[57], C. D. Galvan[81], P. Ganoti[47], C. Garabatos[13], J. R. A. Garcia[82], E. Garcia-Solis[107], K. Garg[31], C. Gargiulo[5], A. Garibli[108], K. Garner[27], P. Gasik[13], E. F. Gauger[67], A. Gautam[109], M. B. Gay Ducati[110], M. Germain[31], P. Ghosh[1], S. K. Ghosh[43], M. Giacalone[20], P. Gianotti[64], P. Giubellino[13,16], P. Giubilato[52], A. M. C. Glaenzer[40], P. Glassel[28], D. J. Q. Goh[111], V. Gonzalez[112], L. H. Gonzalez-Trueba[18], S. Gorbunov[97], M. Gorgon[41], L. Gorlich[66], S. Gotovac[113], V. Grabski[18], L. K. Graczykowski[98], L. Greiner[30], A. Grelli[114], C. Grigoras[5], V. Grigoriev[58], A. Grigoryan[115,155], S. Grigoryan[53,115], O. S. Groettvik[22], F. Grosa[5], J. F. Grosse-Oetringhaus[5], R. Grosso[13], G. G. Guardiano[86], R. Guernane[92], M. Guilbaud[31], K. Gunji[116], T. Gunji[116], A. Gupta[39], R. Gupta[39], S. P. Guzman[82], L. Gyulai[48], M. K. Habib[13], C. Hadjidakis[34], G. Halimoglu[23], H. Hamagaki[111], G. Hamar[48], M. Hamid[17], R. Hannigan[67], M. R. Haque[98,95], A. Harlenderova[13], J. W. Harris[33], A. Harton[107], J. A. Hasenbichler[5], H. Hassan[70], D. Hatzifotiadou[7], P. Hauer[42], L. B. Havener[33], S. Hayashi[116], S. T. Heckel[5], E. Hellbar[23], H. Helstrup[117], T. Herman[65], E. G. Hernandez[82], G. Herrera Corral[118], F. Herrmann[27], K. F. Hetland[117], H. Hillemanns[5], C. Hills[51], B. Hippolyte[57], B. Hofman[114], B. Hohlweger[26,45], J. Honermann[27], G. H. Hong[119], D. Horak[65], S. Hornung[13], A. Horzyk[41], R. Hosokawa[120], P. Hristov[5], C. Hughes[121], P. Huhn[23], T. J. Humanic[122], H. Hushnud[84], L. A. Husova[27], A. Hutson[21], D. Hutter[97], J. P. Iddon[5,51], R. Ilkaev[77], H. Ilyas[19], M. Inaba[87], G. M. Innocenti[5], M. Ippolitov[15], A. Isakov[2,65], M. S. Islam[84], M. Ivanov[13], V. Ivanov[60], V. Izucheev[105], B. Jacak[30], N. Jacazio[5], P. M. Jacobs[30], S. Jadlovska[123], J. Jadlovsky[123], S. Jaelani[114], C. Jahnke[74,86], M. J. Jakubowska[98], A. Jalotra[39], M. A. Janik[98], T. Janson[3], M. Jercic[104], O. Jevons[91], F. Jonas[70,27], P. G. Jones[91], J. M. Jowett[5,13], J. Jung[23], M. Jung[23], A. Junique[5], A. Jusko[49], J. Kaewjai[124], P. Kalinak[60], A. Kalweit[5], V. Kaplin[58], S. Kar[17], A. Karasu Uysal[126], D. Karatovic[104], O. Karavichev[101], T. Karavicheva[101], P. Karczmarczyk[98], E. Karpechev[101], A. Kazantsev[15], U. Kebschull[3], R. Keidel[127],

D. L. D. Keijdener[114], M. Keil[5], B. Ketzer[42], Z. Khabanova[26], A. M. Khan[17], S. Khan[8], A. Khanzadeev[60], Y. Kharlov[105], A. Khatun[8], A. Khuntia[66], B. Kileng[117], B. Kim[71,128], C. Kim[128], D. Kim[119], D. J. Kim[129], E. J. Kim[130], J. Kim[119], J. S. Kim[36], J. Kim[28], J. Kim[119], J. Kim[130], M. Kim[28], S. Kim[131], T. Kim[119], S. Kirsch[23], I. Kisel[97], S. Kiselev[12], A. Kisiel[98], J. P. Kitowski[41], J. L. Klay[132], J. Klein[5], S. Klein[30], C. Klein-Bosing[27], M. Kleiner[23], T. Klemenz[45], A. Kluge[5], A. G. Knospe[21], C. Kobdaj[124], M. K. Kohler[28], T. Kollegger[13], A. Kondratyev[53], N. Kondratyeva[58], E. Kondratyuk[105], J. Konig[23], S. A. Konigstorfer[45], P. J. Konopka[5,41], G. Kornakov[98], S. D. Koryciak[41], L. Koska[123], A. Kotliarov[2], O. Kovalenko[24], V. Kovalenko[24], M. Kowalski[66], I. Kralik[125], A. Kravcakova[10], L. Kreis[13], M. Krivda[91,125], F. Krizek[2], K. Krizkova Gajdosova[65], M. Kroesen[28], M. Kruger[23], E. Kryshen[60], M. Krzewicki[97], V. Kucera[5], C. Kuhn[57], P. G. Kuijer[26], T. Kumaoka[87], D. Kumar[1], L. Kumar[102], N. Kumar[102], S. Kundu[5,95], P. Kurashvili[133], A. Kurepin[101], A. B. Kurepin[101], A. Kuryakin[77], S. Kushpil[2], J. Kvapil[91], M. J. Kweon[71], J. Y. Kwon[71], Y. Kwon[119], S. L. La Pointe[97], P. La Rocca[44], Y. S. Lai[30], A. Lakrathok[124], M. Lamanna[5], R. Langoy[134], K. Lapidus[5], P. Larionov[64], E. Laudi[5], L. Lautner[5,45], R. Lavicka[65], T. Lazareva[24], R. Lea[69,72,83], J. Lehrbach[97], R. C. Lemmon[49], I. Leon Monzon[81], E. D. Lesser[99], M. Lettrich[5,45], P. Levai[48], X. Li[135], X. L. Li[17], J. Lien[134], R. Lietava[91], B. Lim[128], S. H. Lim[128], V. Lindenstruth[97], A. Lindner[25], C. Lippmann[13], A. Liu[99], J. Liu[51], I. M. Lofnes[22], V. Loginov[58], C. Loizides[70], P. Loncar[113], J. A. Lopez[28], X. Lopez[50], E. Lopez Torres[100], J. R. Luhder[27], M. Lunardon[52], G. Luparello[106], Y. G. Ma[14], A. Maevskaya[101], M. Mager[5], T. Mahmoud[42], A. Maire[57], M. Malaev[60], N. M. Malik[39], Q. W. Malik[32], L. Malinina[53,152], D. Mal'Kevich[12], N. Mallick[38], P. Malzacher[13], G. Mandaglio[35,136], V. Manko[15], F. Manso[50], V. Manzari[90], Y. Mao[17], J. Mares[137], G. V. Margagliotti[83], A. Margotti[7], A. Marin[13], C. Markert[67], M. Marquard[23], N. A. Martin[28], P. Martinengo[5], J. L. Martinez[21], M. I. Martinez[82], G. Martinez Garcia[31], S. Masciocchi[13], M. Masera[46], A. Masoni[63], L. Massacrier[34], A. Mastroserio[90,138], A. M. Mathis[45], O. Matonoha[4], P. F. T. Matuoka[74], A. Matyja[66], C. Mayer[66], A. L. Mazuecos[5], F. Mazzaschi[46], M. Mazzilli[5], J. E. Mdhluli[80], A. F. Mechler[23], F. Meddi[140], Y. Melikyan[101], A. Menchaca-Rocha[18], E. Meninno[78,96], A. S. Menon[21], M. Meres[103], S. Mhlanga[79,89], Y. Miake[87], L. Micheletti[16,46], L. C. Migliorin[85], D. L. Mihaylov[45], K. Mikhaylov[12,53], A. N. Mishra[48], D. Miskowiec[13], A. Modak[43], A. P. Mohanty[114], B. Mohanty[95], M. Mohisin Khan[8], Z. Moravcova[56], C. Mordasini[45], D. A. Moreira De Godoy[27], L. A. P. Moreno[82], I. Morozov[101], A. Morsch[5], T. Mrnjavac[5], V. Muccifora[64], E. Mudnic[113], D. Muhlheim[27], S. Muhuri[1], J. D. Mulligan[30], A. Mulliri[63], M. G. Munhoz[74], R. H. Munzer[23], H. Murakami[116], S. Murray[89], L. Musa[5], J. Musinsky[125], J. W. Myrcha[98], B. Naik[54,80], R. Nair[133], B. K. Nandi[54], R. Nania[7], E. Nappi[90], M. U. Naru[19], A. F. Nassirpour[4], A. Nath[28], C. Nattrass[121], A. Neagu[32], L. Nellen[59], S. V. Nesbo[117], G. Neskovic[97], D. Nesterov[24], B. S. Nielsen[56], S. Nikolaev[15], S. Nikulin[15], V. Nikulin[60], F. Noferini[7], S. Noh[141], P. Nomokonov[53], J. Norman[51], N. Novitzky[87], P. Nowakowski[98], A. Nyanin[15], J. Nystrand[22], M. Ogino[111], A. Ohlson[4], V. A. Okorokov[58], J. Oleniacz[98], A. C. Oliveira Da Silva[121], M. H. Oliver[33], A. Onnerstad[129], C. Oppedisano[16], A. Ortiz Velasquez[59], T. Osako[142], A. Oskarsson[4], J. Otwinowski[66], K. Oyama[111], Y. Pachmayer[28], S. Padhan[54], D. Pagano[69,72], G. Paic[59], A. Palasciano[90], J. Pan[112], S. Panebianco[40], P. Pareek[1], J. Park[71], J. E. Parkkila[129], S. P. Pathak[21], R. N. Patra[5,39], B. Paul[63], J. Pazzini[72,69], H. Pei[17], T. Peitzmann[114], X. Peng[17], L. G. Pereira[110], H. Pereira Da Costa[40], D. Peresunko[15], G. M. Perez[100], S. Perrin[40], Y. Pestov[143], V. Petracek[65], M. Petrovici[25], R. P. Pezzi[31,110], S. Piano[16], M. Pikna[103], P. Pillot[31], O. Pinazza[5,7], L. Pinsky[21], C. Pinto[44], S. Pisano[64], M. Ploskon[30], M. Planinic[104], F. Pliquett[23], M. G. Poghosyan[70], B. Polichtchouk[105], S. Politano[6], N. Poljak[104], A. Pop[25], S. Porteboeuf-Houssais[50], J. Porter[30], V. Pozdniakov[53], S. K. Prasad[43], R. Preghenella[7], F. Prino[16], C. A. Pruneau[112], I. Pshenichnov[101], M. Puccio[5], S. Qiu[26], L. Quaglia[46], R. E. Quishpe[21], S. Ragoni[49], A. Rakotozafindrabe[40], L. Ramello[93], F. Rami[57], S. A. R. Ramirez[82], A. G. T. Ramos[76], T. A. Rancien[92], R. Raniwala[144], S. Raniwala[144], S. S. Rasanen[145], R. Rath[38], I. Ravasenga[26], K. F. Read[70,121], A. R. Redelbach[97], K. Redlich[133,153], A. Rehman[22], P. Reichelt[23], F. Reidt[5], H. A. Reme-ness[117], R. Renfordt[23], Z. Rescakova[10], K. Reygers[28], A. Riabov[60], V. Riabov[60], T. Richert[4,56], M. Richter[32], W. Riegler[5], F. Riggi[44], C. Ristea[61], S. P. Rode[38], M. Rodriguez Cahuantzi[82], K. Roed[32], R. Rogalev[105], E. Rogochaya[53], T. S. Rogoschinski[23], D. Rohr[5], D. Rohrich[22], P. F. Rojas[82], P. S. Rokita[98], F. Ronchetti[64], A. Rosano[35,136], E. D. Rosas[59], A. Rossi[29], A. Rotondi[68,69], A. Roy[38], P. Roy[84], S. Roy[54], N. Rubini[20], O. V. Rueda[4], R. Rui[83], B. Rumyantsev[53], P. G. Russek[41], A. Rustamov[108], E. Ryabinkin[15], Y. Ryabov[60], A. Rybicki[66], H. Rytkonen[129], W. Rzesa[98], O. A. M. Saarimaki[145], R. Sadek[31], S. Sadovsky[105], J. Saetre[22], K. Safarik[65], S. K. Saha[1], S. Saha[95], B. Sahoo[54], P. Sahoo[54], R. Sahoo[38], S. Sahoo[38], D. Sahu[38], P. K. Sahu[54], J. Saini[1], S. Sakai[87], S. Sambyal[39], V. Samsonov[58,60,157], D. Sarkar[112], N. Sarkar[1], P. Sarma[62], V. M. Sarti[45], M. H. P. Sas[33], J. Schambach[70,67], H. S. Scheid[23], C. Schiaua[25], R. Schicker[28], A. Schmah[28], C. Schmidt[13], H. R. Schmidt[147], M. O. Schmidt[28], M. Schmidt[147], N. V. Schmidt[23,70], A. R. Schmier[121], R. Schotter[57], J. Schukraft[5], Y. Schutz[57], K. Schwarz[13], K. Schweda[13], G. Scioli[20], E. Scomparin[16], J. E. Seger[120], Y. Sekiguchi[116], D. Sekihata[116], I. Selyuzhenkov[13,58], S. Senyukov[57], J. J. Seo[71], D. Serebryakov[101], L. Serksnyte[45], A. Sevcenco[61], T. J. Shaba[79], A. Shabanov[101], A. Shabetai[31], R. Shahoyan[5], W. Shaikh[84], A. Shangaraev[105], A. Sharma[102], H. Sharma[66], M. Sharma[39], N. Sharma[102], S. Sharma[39], U. Sharma[39], O. Sheibani[21], K. Shigaki[142], M. Shimomura[148], S. Shirinkin[12], Q. Shou[14], Y. Sibiriak[15], S. Siddhanta[88], T. Siemiarczuk[133], T. F. Silva[74], D. Silvermyr[4], G. Simonetti[5], B. Singh[45], R. Singh[95], R. Singh[39], R. Singh[38], V. K. Singh[1], V. Singhal[1], T. Sinha[84], B. Sitar[103], M. Sitta[93], T. B. Skaali[32], G. Skorodumovs[28], M. Slupecki[145], N. Smirnov[33], R. J. M. Snellings[114], C. Soncco[55], J. Song[21], A. Songmoolnak[124], F. Soramel[52], S. Sorensen[121], I. Sputowska[66], J. Stachel[28], I. Stan[61], P. J. Steffanic[121], S. F. Stiefelmaier[28], D. Stocco[31], I. Storehaug[32], M. M. Storetvedt[117], C. P. Stylianidis[26], A. A. P. Suaide[74], T. Sugitate[142], C. Suire[34], M. Suljic[5], R. Sultanov[12], M. Sumbera[2], V. Sumberia[39], S. Sumowidagdo[11], S. Swain[54], A. Szabo[103], I. Szarka[103], U. Tabassam[19], S. F. Taghavi[45], G. Taillepied[50], J. Takahashi[86], G. J. Tambave[22], S. Tang[37,50], Z. Tang[37], M. Tarhini[31], M. G. Tarzila[25], A. Tauro[5], G. Tejeda Munoz[82], A. Telesca[5], L. Terlizzi[46], C. Terrevoli[21], G. Tersimonov[149], S. Thakur[1], D. Thomas[67], R. Tieulent[85], A. Tikhonov[101], A. R. Timmins[21], M. Tkacik[123], A. Toia[23], N. Topilskaya[101], M. Toppi[64], F. Torales-Acosta[99], T. Tork[34], S. R. Torres[35,136], A. Trifiro[35,136], S. Tripathy[7,59], T. Tripathy[54], S. Trogolo[5], G. Trombetta[76], V. Trubnikov[149], W. H. Trzaska[129], T. P. Trzcinski[98], B. A. Trzeciak[65], A. Tumkin[77], R. Turrisi[29], T. S. Tveter[32], K. Ullaland[22], A. Uras[85], M. Urioni[69,72], G. L. Usai[63], M. Vala[10], N. Valle[68,69], S. Vallero[16], N. van der Kolk[114], L. V. R. van Doremalen[114], M. van Leeuwen[26], P. Vande Vyvre[5], D. Varga[48], Z. Varga[48], M. Varga-Kofarago[48], A. Vargas[82], M. Vasileiou[47], A. Vasiliev[15], O. Vazquez Doce[64], V. Vechernin[24], E. Vercellin[46], S. Vergara Limon[82], L. Vermunt[114], R. Vertesi[48], M. Verweij[114], L. Vickovic[113], Z. Vilakazi[80], O. Villalobos Baillie[91], G. Vino[90], A. Vinogradov[15], T. Virgili[96], V. Vislavicius[56], A. Vodopyanov[53], B. Volkel[5],

M. A. Volkl[28], K. Voloshin[12], S. A. Voloshin[112], G. Volpe[76], B. von Haller[5], I. Vorobyev[45], D. Voscek[123], N. Vozniuk[101], J. Vrlakova[10], B. Wagner[22], C. Wang[14], D. Wang[14], M. Weber[78], R. J. G. V. Weelden[26], A. Wegrzynek[5], S. C. Wenzel[5], J. P. Wessels[27], J. Wiechula[23], J. Wikne[32], G. Wilk[133], J. Wilkinson[13], G. A. Willems[27], B. Windelband[28], M. Winn[40], W. E. Witt[121], J. R. Wright[67], W. Wu[14], Y. Wu[37], R. Xu[17], S. Yalcin[126], Y. Yamaguchi[142], K. Yamakawa[142], S. Yang[22], S. Yano[142], Z. Yin[17], H. Yokoyama[114], I.-K. Yoo[128], J. H. Yoon[71], S. Yuan[22], A. Yuncu[28], V. Zaccolo[83], A. Zaman[19], C. Zampolli[5], H. J. C. Zanoli[114], N. Zardoshti[5], A. Zarochentsev[24], P. Zavada[137], N. Zaviyalov[77], H. Zbroszczyk[98], M. Zhalov[60], S. Zhang[14], X. Zhang[14], Y. Zhang[37], V. Zherebchevskii[24], Y. Zhi[135], N. Zhigareva[12], D. Zhou[17], Y. Zhou[56], J. Zhu[13,17], Y. Zhu[17], A. Zichichi[20], G. Zinovjev[149] & N. Zurlo[69,72]

[1]Variable Energy Cyclotron Centre, Homi Bhabha National Institute, Kolkata, India. [2]Nuclear Physics Institute of the Czech Academy of Sciences, Řežu Prahy, Czech Republic. [3]Institut fur Informatik, Fachbereich Informatik und Mathematik, Johann-Wolfgang-Goethe Universitat Frankfurt, Frankfurt, Germany. [4]Department of Physics, Division of Particle Physics, Lund University, Lund, Sweden. [5]European Organization for Nuclear Research (CERN), Geneva, Switzerland. [6]Dipartimento DISAT del Politecnico and Sezione INFN, Turin, Italy. [7]INFN, Sezione di Bologna, Bologna, Italy. [8]Department of Physics, Aligarh Muslim University, Aligarh, India. [9]Korea Institute of Science and Technology Information, Daejeon, Republic of Korea. [10]Faculty of Science, P.J. Šafarik University, Košice, Slovakia. [11]Indonesian Institute of Sciences, Jakarta, Indonesia. [12]NRC «Kurchatov» Institute – ITEP, Moscow, Russia. [13]Research Division and ExtreMe Matter Institute EMMI, GSI Helmholtzzentrum fur Schwerionenforschung GmbH, Darmstadt, Germany. [14]Fudan University, Shanghai, China. [15]National Research Centre Kurchatov Institute, Moscow, Russia. [16]INFN, Sezione di Torino, Turin, Italy. [17]Central China Normal University, Wuhan, China. [18]Instituto de Fisica, Universidad Nacional Autonoma de Mexico, Mexico City, Mexico. [19]COMSATS University Islamabad, Islamabad, Pakistan. [20]Dipartimento di Fisica e Astronomia dell'Universita and Sezione INFN, Bologna, Italy. [21]University of Houston, Houston, TX, USA. [22]Department of Physics and Technology, University of Bergen, Bergen, Norway. [23]Institut fur Kernphysik, Johann Wolfgang Goethe-Universitat Frankfurt, Frankfurt, Germany. [24]St. Petersburg State University, St. Petersburg, Russia. [25]Horia Hulubei National Institute of Physics and Nuclear Engineering, Bucharest, Romania. [26]Nikhef, National Institute for Subatomic Physics, Amsterdam, the Netherlands. [27]Institut fur Kernphysik, Westfalische Wilhelms-Universitat Munster, Munster, Germany. [28]Physikalisches Institut, Ruprecht-Karls-Universitat Heidelberg, Heidelberg, Germany. [29]INFN, Sezione di Padova, Padova, Italy. [30]Lawrence Berkeley National Laboratory, Berkeley, CA, USA. [31]SUBATECH, IMT Atlantique, Universite de Nantes, CNRS-IN2P3, Nantes, France. [32]Department of Physics, University of Oslo, Oslo, Norway. [33]Yale University, New Haven, CT, USA. [34]Laboratoire de Physique des 2 Infinis, Irene Joliot-Curie, Orsay, France. [35]INFN, Sezione di Catania, Catania, Italy. [36]Gangneung-Wonju National University, Gangneung, Republic of Korea. [37]University of Science and Technology of China, Hefei, China. [38]Indian Institute of Technology Indore, Indore, India. [39]Physics Department, University of Jammu, Jammu, India. [40]Department de Physique Nucleaire (DPhN), Universite Paris-Saclay Centre d'Etudes de Saclay (CEA), IRFU, Saclay, France. [41]AGH University of Science and Technology, Cracow, Poland. [42]Helmholtz-Institut fur Strahlen- und Kernphysik, Rheinische Friedrich-Wilhelms-Universitat Bonn, Bonn, Germany. [43]Bose Institute, Department of Physics and Centre for Astroparticle Physics and Space Science (CAPSS), Kolkata, India. [44]Dipartimento di Fisica e Astronomia dell'Universita and Sezione INFN, Catania, Italy. [45]Physik Department, Technische Universitat Munchen, Munich, Germany. [46]Dipartimento di Fisica dell'Universita and Sezione INFN, Turin, Italy. [47]Department of Physics, School of Science, National and Kapodistrian University of Athens,, Athens, Greece. [48]Wigner Research Centre for Physics, Budapest, Hungary. [49]Nuclear Physics Group, STFC Daresbury Laboratory, Daresbury, UK. [50]Universite Clermont Auvergne, CNRS/IN2P3, LPC, Clermont-Ferrand, France. [51]University of Liverpool, Liverpool, UK. [52]Dipartimento di Fisica e Astronomia dell'Universita and Sezione INFN, Padova, Italy. [53]Joint Institute for Nuclear Research (JINR), Dubna, Russia. [54]Indian Institute of Technology Bombay (IIT), Mumbai, India. [55]Seccion Fisica, Departamento de Ciencias, Pontificia Universidad Catolica del Peru, Lima, Peru. [56]Niels Bohr Institute, University of Copenhagen, Copenhagen, Denmark. [57]Universite de Strasbourg, CNRS, IPHC UMR 7178, Strasbourg, France. [58]NRNU Moscow Engineering Physics Institute, Moscow, Russia. [59]Instituto de Ciencias Nucleares, Universidad Nacional Autonoma de Mexico, Mexico City, Mexico. [60]Petersburg Nuclear Physics Institute, Gatchina, Russia. [61]Institute of Space Science (ISS), Bucharest, Romania. [62]Department of Physics, Gauhati University, Guwahati, India. [63]Dipartimento di Fisica dell'Universita and Sezione INFN, Cagliari, Italy. [64]INFN, Laboratori Nazionali di Frascati, Frascati, Italy. [65]Faculty of Nuclear Sciences and Physical Engineering, Czech Technical University in Prague, Prague, Czech Republic. [66]The Henryk Niewodniczanski Institute of Nuclear Physics, Polish Academy of Sciences, Cracow, Poland. [67]The University of Texas at Austin, Austin, TX, USA. [68]Dipartimento di Fisica e Nucleare e Teorica, Universita di Pavia, Pavia, Italy. [69]INFN, Sezione di Pavia, Pavia, Italy. [70]Oak Ridge National Laboratory, Oak Ridge, TN, USA. [71]Inha University, Incheon, Republic of Korea. [72]Universita di Brescia, Brescia, Italy. [73]Moscow Institute for Physics and Technology, Moscow, Russia. [74]Universidade de Sao Paulo (USP), Sao Paulo, Brazil. [75]Politecnico di Bari and Sezione INFN, Bari, Italy. [76]Dipartimento Interateneo di Fisica 'M. Merlin' and Sezione INFN, Bari, Italy. [77]Russian Federal Nuclear Center (VNIIEF), Sarov, Russia. [78]Stefan Meyer Institut fur Subatomare Physik (SMI), Vienna, Austria. [79]iThemba LABS, National Research Foundation, Somerset West, South Africa. [80]University of the Witwatersrand, Johannesburg, South Africa. [81]Universidad Autonoma de Sinaloa, Culiacan, Mexico. [82]High Energy Physics Group, Universidad Autonoma de Puebla, Puebla, Mexico. [83]Dipartimento di Fisica dell'Universita and Sezione INFN, Trieste, Italy. [84]Saha Institute of Nuclear Physics, Homi Bhabha National Institute, Kolkata, India. [85]Universite de Lyon, CNRS/IN2P3, Institut de Physique des 2 Infinis de Lyon, Lyon, France. [86]Universidade Estadual de Campinas (UNICAMP), Campinas, Brazil. [87]University of Tsukuba, Tsukuba, Japan. [88]INFN, Sezione di Cagliari, Cagliari, Italy. [89]University of Cape Town, Cape Town, South Africa. [90]INFN, Sezione di Bari, Bari, Italy. [91]School of Physics and Astronomy, University of Birmingham, Birmingham, UK. [92]Laboratoire de Physique Subatomique et de Cosmologie, Universite Grenoble-Alpes, CNRS-IN2P3, Grenoble, France. [93]Dipartimento di Scienze e Innovazione Tecnologica dell'Universita del Piemonte Orientale and INFN Sezione di Torino, Alessandria, Italy. [94]Universidade Federal do ABC, Santo Andre, Brazil. [95]National Institute of Science Education and Research, Homi Bhabha National Institute, Jatni, India. [96]Dipartimento di Fisica 'E.R. Caianiello' dell'Universita and Gruppo Collegato INFN, Salerno, Italy. [97]Frankfurt Institute for Advanced Studies, Johann Wolfgang Goethe-Universitat Frankfurt, Frankfurt, Germany. [98]Warsaw University of Technology, Warsaw, Poland. [99]Department of Physics, University of California, Berkeley, CA, USA. [100]Centro de Aplicaciones Tecnologicas y Desarrollo Nuclear (CEADEN), Havana, Cuba. [101]Institute for Nuclear Research, Academy of Sciences, Moscow, Russia. [102]Physics Department, Panjab University, Chandigarh, India. [103]Comenius University Bratislava, Faculty of Mathematics, Physics and Informatics, Bratislava, Slovakia. [104]Physics Department, Faculty of Science, University of Zagreb, Zagreb, Croatia. [105]NRC Kurchatov Institute IHEP, Protvino, Russia. [106]INFN, Sezione di Trieste, Trieste, Italy. [107]Chicago State University, Chicago, IL, USA. [108]National Nuclear Research Center, Baku, Azerbaijan. [109]University of Kansas, Lawrence, KS, USA. [110]Instituto de Fisica, Universidade Federal do Rio Grande do Sul (UFRGS), Porto Alegre, Brazil. [111]Nagasaki Institute of Applied Science, Nagasaki, Japan. [112]Wayne State University, Detroit, MI, USA. [113]Faculty of Electrical Engineering, Mechanical Engineering and Naval Architecture, University of Split, Split, Croatia. [114]Institute for Gravitational and Subatomic Physics (GRASP), Utrecht University/Nikhef, Utrecht, the Netherlands. [115]A.I. Alikhanyan National Science Laboratory (Yerevan Physics Institute) Foundation, Yerevan, Armenia. [116]University of Tokyo, Tokyo, Japan. [117]Faculty of Engineering and Science, Western Norway University of Applied Sciences, Bergen, Norway. [118]Centro de Investigacion y de Estudios Avanzados (CINVESTAV), Mexico City and Merida, Mexico. [119]Yonsei University, Seoul, Republic of Korea. [120]Creighton University, Omaha, NE, USA. [121]University of Tennessee, Knoxville, TN, USA. [122]Ohio State University, Columbus, OH, USA. [123]Technical University of Košice, Košice, Slovakia. [124]Suranaree University of Technology, Nakhon Ratchasima, Thailand. [125]Institute of Experimental Physics, Slovak Academy of Sciences, Košice, Slovakia. [126]KTO Karatay University, Konya, Turkey. [127]Hochschule Worms, Zentrum fur Technologietransfer und Telekommunikation (ZTT), Worms, Germany. [128]Department of Physics, Pusan National University, Pusan, Republic of Korea. [129]University of Jyvaskyla, Jyvaskyla, Finland. [130]Jeonbuk National University, Jeonju, Republic of Korea. [131]Department of Physics, Sejong University, Seoul, Republic of Korea. [132]California Polytechnic State University, San Luis Obispo, CA, USA. [133]National Centre for Nuclear Research, Warsaw, Poland. [134]University of South-Eastern Norway, Tonsberg, Norway. [135]China Institute of Atomic Energy, Beijing, China. [136]Dipartimento di Scienze MIFT, Universita di Messina, Messina, Italy. [137]Institute of Physics of the Czech Academy of Sciences, Prague, Czech Republic. [138]Universita degli Studi di Foggia, Foggia, Italy. [139]INFN, Sezione di Roma, Rome, Italy. [140]Dipartimento di Fisica dell'Universita 'La Sapienza' and Sezione INFN, Rome, Italy. [141]Chungbuk National University, Cheongju, Republic of Korea. [142]Hiroshima University, Hiroshima, Japan. [143]Budker Institute for Nuclear Physics, Novosibirsk, Russia. [144]Physics Department, University of Rajasthan, Jaipur, India. [145]Helsinki Institute of Physics (HIP), Helsinki, Finland. [146]Institute of Physics, Homi Bhabha National Institute, Bhubaneswar, India. [147]Physikalisches Institut, Eberhard-Karls-Universitat Tubingen, Tubingen, Germany. [148]Nara Women's University (NWU), Nara, Japan. [149]Bogolyubov Institute for Theoretical Physics, National Academy of Sciences of Ukraine, Kiev, Ukraine. [150]Present address: Italian National Agency for New Technologies, Energy and Sustainable Economic Development, (ENEA), Bologna, Italy. [151]Present address: Dipartimento DET del Politecnico di Torino, Turin, Italy. [152]Present address: D.V. Skobeltsyn Institute of Nuclear Physics, M.V. Lomonosov Moscow State University, Moscow, Russia. [153]Present address: Institute of Theoretical Physics, University of Wroclaw, Wroclaw, Poland. [154]Deceased: J. Cleymans. [155]Deceased: A. Grigoryan. [156]Deceased: M. A. Mazzoni. [157]Deceased: V. Samsonov.

## Methods

### Detector setup and data set

The analysis was performed with the ALICE detector at the CERN LHC[41]. The ALICE Inner Tracking System[42] and Time Projection Chamber[43] were used for charged-particle reconstruction, and particle identification (PID) was obtained using the combined information from the Time Projection Chamber and the Time-Of-Flight detectors[44]. These detectors are located in the ALICE central barrel, which has full azimuthal coverage and a pseudorapidity range of $|\eta| < 0.9$. The data set used in this analysis was collected in 2016, 2017 and 2018 in proton–proton collisions at $\sqrt{s} = 13$ TeV, with a minimum-bias trigger condition defined by the presence of at least one hit in each of the two V0 scintillators[45]. This trigger accepts all events of interest for this analysis and the collected data sample corresponds to an integrated luminosity of $\mathcal{L}_{int} = 25$ nb$^{-1}$.

### Jet finding and tagging

Jet finding was performed using the anti-$k_T$ algorithm, with a jet resolution parameter of $R = 0.4$. The $E$-scheme recombination strategy was chosen to combine the tracks of the jet by adding their four-momenta, with a geometric constraint on the pseudorapidity of $|\eta| < 0.5$ enforced on the jet axis, to ensure that the full jet cone was contained in the acceptance of the central barrel of the ALICE detector. The ALICE detector has excellent tracking efficiency down to low $p_T$ (approximately 80% at $p_T = 500$ MeV/$c$), which is homogeneous as a function of pseudorapidity and azimuthal angle[41], within the acceptance. The effect of track density on the tracking efficiency is also negligible[46]. The angular resolution is about 20% down to splitting angles of 0.05 radians, which motivated a track-based jet measurement as opposed to a full jet measurement using calorimetric information. Recent measurements[15,20] have shown that track-based jet observables are successful at reconstructing the parton shower information through declustering techniques, despite missing the information from the neutral component of the jet.

Jets with a transverse momentum in the interval of $5 \leq p_{T,jet}^{ch} < 50$ GeV/$c$ were selected for this analysis. To mitigate against the cases in which two $D^0$-meson candidates share a common decay track, jet-finding passes were performed independently for each $D^0$-meson candidate in the event, each time replacing only the decay tracks of that candidate with the corresponding $D^0$-meson candidate. In each pass the jet containing the reconstructed $D^0$-meson candidate of that pass was subsequently tagged as a charm-initiated jet candidate.

### Subtraction of the combinatorial background in the $D^0$-meson candidate sample

To extract the true $D^0$-meson tagged jet $R(\theta)$ distributions and remove the contribution from combinatorial $K^\mp \pi^\pm$ pairs surviving the topological and PID selections, a side-band subtraction procedure was used. This involved dividing the sample into $p_T^{D^0}$ intervals and fitting the invariant-mass distributions of the $D^0$ candidates in each interval with a Gaussian function for the signal and an exponential function for the background. The width ($\sigma$) and mean of the fitted Gaussian were used to define signal and side-band regions, with the two-dimensional distributions of $\theta$ and $E_{Radiator}$ for $D^0$-meson tagged jet candidates, $\rho(\theta, E_{Radiator})^{D^0 \text{ jet candidate}}$, obtained in each region. The signal region was defined to be within $2\sigma$ on either side of the Gaussian mean and contained most of the real $D^0$ mesons, with some contamination present from the combinatorial background. The side-band regions were defined to be from $4\sigma$ to $9\sigma$ away from the peak in either direction and were composed entirely of background $D^0$-meson candidates. The combined $\rho(\theta, E_{Radiator})^{D^0 \text{ jet candidate}}$ distributions measured in the two side-band regions represent the structural form of the contribution of background candidates to the $\rho(\theta, E_{Radiator})^{D^0 \text{ jet candidate}}$ distribution measured in the signal region. In this way, the background component of the total $\rho(\theta, E_{Radiator})^{D^0 \text{ jet candidate}}$ measured in the signal region can be subtracted, using the following equation:

$$\rho(\theta, E_{Radiator})^{D^0 \text{ jet}}$$
$$= \sum_i \frac{1}{\varepsilon_i} [(\rho(\theta, E_{Radiator})_S^{D^0 \text{ jet candidate}} - \frac{A_S}{A_B}\rho(\theta, E_{Radiator})_B^{D^0 \text{ jet candidate}}] \quad (3)$$

where the subscripts S and B denote the signal and side-band regions of the invariant-mass distributions, respectively. The $A_S$ and $A_B$ variables are the areas under the background fit function in the signal and combined side-band regions, respectively, and were used to normalize the magnitude of the background in the side-band regions to that in the signal region. The $D^0$-meson tagged jet selection efficiency (discussed in more detail in the next section) is denoted by $\varepsilon$, with the index $i$ running over the $p_T^{D^0}$ bins. As a result of this side-band subtraction, the true $D^0$-meson tagged jet $\rho(\theta, E_{Radiator})^{D^0 \text{ jet}}$ distributions are obtained, in the different intervals of $p_T^{D^0}$.

### $D^0$-meson tagged jet reconstruction efficiency correction

The topological and PID selections used to identify the $D^0$ mesons, in the chosen jet kinematic interval, have a limited efficiency, which exhibits a strong $p_T$ dependence. Therefore, before integrating the side-band subtracted $\rho(\theta, E_{Radiator})^{D^0 \text{ jet}}$ distributions across the measured $p_T^{D^0}$ intervals, the $\rho(\theta, E_{Radiator})^{D^0 \text{ jet}}$ distributions were corrected for this efficiency. The efficiency, $\varepsilon$, was estimated from PYTHIA v.6 MC studies and varies strongly with $p_T^{D^0}$, from approximately 0.01 at $p_T^{D^0} = 2.5$ GeV/$c$ to approximately 0.3 at $p_T^{D^0} = 30$ GeV/$c$ for prompt $D^0$-meson tagged jets and from approximately 0.01 at $p_T^{D^0} = 2.5$ GeV/$c$ to approximately 0.2 at $p_T^{D^0} = 30$ GeV/$c$ for non-prompt $D^0$-meson tagged jets. As the prompt and non-prompt $D^0$-meson tagged jet reconstruction efficiencies were different, the final efficiency was obtained by combining the prompt and non-prompt $D^0$-meson tagged jet reconstruction efficiencies, evaluated separately. These were combined with weights derived from simulations, corresponding to the admixture of prompt and non-prompt $D^0$-meson tagged jets in the reconstructed sample. The fractions of this admixture were obtained in bins of $p_T^{D^0}$ by calculating the prompt and non-prompt $D^0$-meson tagged jet production cross sections with POWHEG combined with PYTHIA v.6 showering.

### Evaluation of systematic uncertainties

Considered sources of systematic uncertainty in the measurement relate to the reconstruction and signal extraction of $D^0$-meson candidates, with the former contributing as the leading source. These uncertainties were estimated by varying the topological and PID selections, as well as the fitting and side-band subtraction configurations applied to the $D^0$-meson candidate invariant mass distributions. Variations were chosen that tested the influence of selected analysis parameters as much as possible, while maintaining a reasonable significance in the signal extraction. For each of these categories, the root mean square of all deviations was taken as the final systematic uncertainty. Theoretical uncertainties in the prompt and non-prompt $D^0$-meson tagged jet production cross sections from POWHEG were also considered in the calculation of the reconstruction efficiency, with the largest variation taken as the uncertainty. For each category, the final systematic uncertainty was symmetrized before adding up the uncertainties in quadrature across all categories to obtain the total systematic uncertainty of the $D^0$-meson tagged jet measurement.

For the inclusive jet results, the minimum $p_T$ requirement on the track with the highest transverse momentum within the leading prong of each splitting was varied. The magnitude of the variation was taken to be the resolution of the transverse momentum of a $D^0$ meson with $p_T^{D^0} = 2$ GeV/$c$, which was found to be 0.06 GeV/$c$. Variations above and below the nominal selection value were made and the largest deviation was symmetrized. Systematic detector effects are dominated by the tracking efficiency and were shown in detector simulations to affect both the $D^0$-meson tagged jet and inclusive jet samples equally, and

they largely cancelled in the $R(\theta)$ ratio. Therefore, the systematic uncertainty of $R(\theta)$ because of detector effects was estimated directly on the ratio by randomly removing 15% of the reconstructed tracks, as given by the tracking efficiency of the ALICE detector, in the track samples used for clustering both the $D^0$-meson tagged jets and inclusive jets. The ratio of the resulting $R(\theta)$ distribution to the case with no track removal was taken, to obtain the uncertainty, which was symmetrized.

The relative uncertainty of $R(\theta)$ resulting from the separate $D^0$-meson tagged jet and inclusive jet uncertainties was calculated, with the resulting absolute uncertainty added in quadrature to the detector effects uncertainty to obtain the total systematic uncertainty of the $R(\theta)$ measurement. The magnitude of each of these sources of systematic uncertainty is shown in Table 1, for the smallest-angle splittings corresponding to the interval $2 \le \ln(1/\theta) < 3$, in which the uncertainties are largest.

**Table 1 | $R(\theta)$ systematic uncertainties**

| Source | $E_{Radiator}$ | | |
|---|---|---|---|
| | 5–10 GeV | 10–20 GeV | 20–35 GeV |
| Invariant-mass fitting | 2.3 | 1.4 | 3.0 |
| Side-band subtraction | 2.0 | 1.8 | 1.4 |
| $D^0$-jet selection stability | 4.1 | 5.0 | 7.2 |
| Non-prompt contribution | 1.0 | 3.5 | 1.1 |
| Leading hadron $p_T$ selection | 2.0 | 3.2 | 0.2 |
| Detector effects | 0.7 | 5.2 | 0.9 |
| Total | 5.6 | 8.9 | 8.1 |

The percentage magnitude of the systematic uncertainties of each source considered, and the total systematic uncertainty, for the $R(\theta)$ variable are shown for the smallest splitting-angle interval $2 \le \ln(1/\theta) < 3$.

## Data availability

All data shown in histograms and plots are publicly available on the HEPdata repository (https://hepdata.net).

## Code availability

The source code utilized in this study is publicly available under the names AliPhysics and AliRoot. Further information can be provided by the authors upon reasonable request.

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

**Acknowledgements** The ALICE Collaboration is grateful to U. Wiedemann for his valuable suggestions and fruitful discussions. We are also grateful to D. Napoletano for providing the SHERPA configuration files and useful discussions. The ALICE Collaboration would like to thank all its engineers and technicians for their invaluable contributions to the construction of the experiment and the CERN accelerator teams for the outstanding performance of the LHC complex. The ALICE Collaboration gratefully acknowledges the resources and support provided by all Grid centres and the Worldwide LHC Computing Grid (WLCG) collaboration. The ALICE Collaboration acknowledges the following funding agencies for their support in building and running the ALICE detector: A. I. Alikhanyan National Science Laboratory (Yerevan Physics Institute) Foundation (ANSL), State Committee of Science and World Federation of Scientists (WFS), Armenia; Austrian Academy of Sciences, Austrian Science Fund (FWF) (M 2467-N36) and Nationalstiftung für Forschung, Technologie und Entwicklung, Austria; Ministry of Communications and High Technologies, National Nuclear Research Center, Azerbaijan; Conselho Nacional de Desenvolvimento Científico e Tecnológico (CNPq), Financiadora de Estudos e Projetos (Finep), Fundação de Amparo à Pesquisa do Estado de São Paulo (FAPESP) and Universidade Federal do Rio Grande do Sul (UFRGS), Brazil; Ministry of Education of China (MOEC), Ministry of Science & Technology of China (MSTC) and National Natural Science Foundation of China (NSFC), China; Ministry of Science and Education and Croatian Science Foundation, Croatia; Centro de Aplicaciones Tecnológicas y Desarrollo Nuclear (CEADEN), Cubaenergía, Cuba; Ministry of Education, Youth and Sports of the Czech Republic, Czech Republic; The Danish Council for Independent Research | Natural Sciences, the VILLUM FONDEN and Danish National Research Foundation (DNRF), Denmark; Helsinki Institute of Physics (HIP), Finland; Commissariat à l'Energie Atomique (CEA), Institut National de Physique Nucléaire et de Physique des Particules (IN2P3) and Centre National de la Recherche Scientifique (CNRS), France; Bundesministerium für Bildung und Forschung (BMBF) and GSI Helmholtzzentrum für Schwerionenforschung GmbH, Germany; General Secretariat for Research and Technology, Ministry of Education, Research and Religions, Greece; National Research, Development and Innovation Office, Hungary; Department of Atomic Energy Government of India (DAE), Department of Science and Technology, Government of India (DST), University Grants Commission, Government of India (UGC) and Council of Scientific and Industrial Research (CSIR), India; Indonesian Institute of Science, Indonesia; Istituto Nazionale di Fisica Nucleare (INFN), Italy; Institute for Innovative Science and Technology, Nagasaki Institute of Applied Science (IIST), Japanese Ministry of Education, Culture, Sports, Science and Technology (MEXT) and Japan Society for the Promotion of Science (JSPS) KAKENHI, Japan; Consejo Nacional de Ciencia (CONACYT) y Tecnología, through Fondo de Cooperación Internacional en Ciencia y Tecnología (FONCICYT) and Dirección General de Asuntos del Personal Academico (DGAPA), Mexico; Nederlandse Organisatie voor Wetenschappelijk Onderzoek (NWO), the Netherlands; The Research Council of Norway, Norway; Commission on Science and Technology for Sustainable Development in the South (COMSATS), Pakistan; Pontificia Universidad Católica del Perú, Peru; Ministry of Education and Science, National Science Centre and WUT ID-UB, Poland; Korea Institute of Science and Technology Information and National Research Foundation of Korea (NRF), Republic of Korea; Ministry of Education and Scientific Research, Institute of Atomic Physics and Ministry of Research and Innovation and Institute of Atomic Physics, Romania; Joint Institute for Nuclear Research (JINR), Ministry of Education and Science of the Russian Federation, National Research Centre Kurchatov Institute, Russian Science Foundation and Russian Foundation for Basic Research, Russia; Ministry of Education, Science, Research and Sport of the Slovak Republic, Slovakia; National Research Foundation of South Africa, South Africa; Swedish Research Council (VR) and Knut & Alice Wallenberg Foundation (KAW), Sweden; European Organization for Nuclear Research, Switzerland; Suranaree University of Technology (SUT), National Science and Technology Development Agency (NSDTA) and Office of the Higher Education Commission under NRU project of Thailand, Thailand; Turkish Energy, Nuclear and Mineral Research Agency (TENMAK), Turkey; National Academy of Sciences of Ukraine, Ukraine; Science and Technology Facilities Council (STFC), United Kingdom; National Science Foundation of the United States of America (NSF) and United States Department of Energy, Office of Nuclear Physics (DOE NP), United States of America.

**Author contributions** All authors have contributed to the publication, being variously involved in the design and the construction of the detectors, in writing software, calibrating subsystems, operating the detectors and acquiring data, and finally analysing the processed data. The ALICE Collaboration members discussed and approved the scientific results. The manuscript was prepared by a subgroup of authors appointed by the collaboration and subject to an internal collaboration-wide review process. All authors reviewed and approved the final version of the manuscript.

**Competing interests** The authors declare no competing interests.

**Additional information**
**Correspondence and requests for materials** should be addressed to the ALICE Collaboration.
