## [Peer Review File · Nature]

Manuscript Title: Direct observation of the dead-cone effect in QCD

Reviewer Comments & Author Rebuttals

Reviewer Reports on the Initial Version:

Referees' comments:

Referee #1 (Remarks to the Author):

Report on "Direct observation of the dead-cone effect in QCD" by the ALICE collaboration.

This manuscript reports the direct observation of an effect in Quantum Chromodynamics (QCD) that was predicted long ago, but had yet to be decisively observed. That effect concerns the bremsstrahlung radiation from a fast-moving quark, and its partial suppression for massive quarks, specifically within angles (relative to the quark) of the order of the ratio of the quark rest mass to the quark energy.

The experimental analysis is clearly explained, and the final result in figure 2 is convincing, especially for lower quark ("radiator") energies. The study is backed up by an estimate of the statistical significance of the effect, which comfortably passes the threshold of 5 standard deviations that is commonly used in particle physics to claim "observation" of a new particle or phenomenon.

Given the importance of the phenomenon, a cornerstone of QCD, and the high quality of the analysis, I believe that the result meets the very high threshold required for publication in Nature. Before publication, however, some points need to be addressed / corrected in the manuscript's presentation of the general context, as outlined below.

Points to address:

- 1) Both the abstract and introduction make the statement that the "vacuum is not transparent to the partons and induces gluon radiation [...]". This is incorrect: it would be like saying that the vacuum is not transparent to electrons and that this induces photon radiation, which is patently false (non-transparency would imply that a free electron travelling through the vacuum would progressively lose energy and eventually come to a stop, which does not happen). Rather, the gluon radiation is the analogue of Bremsstrahlung in QED and is induced by the change in direction (acceleration) of the parton when it scatters off another parton. These statements need to be corrected before the manuscript can be published.
- 2) The statement, in the introduction that "the corresponding fraction carried by the leading hadron in light quark-initiated jets peaks at values close to zero" is correct only for case of extremely-energetic quark-initiated jets. For the energies being studied by the ALICE collaboration, the peak momentum fraction carried by the leading hadron is probably around 0.25 (the authors may want to verify this with a simulation, or simply drop the corresponding part of the sentence).
- 3) Still in the introduction, the statement that "the dead-cone angular region can be filled by hadronisation effects" is, I think incorrect. The dead cone is not a total absence of radiation, but rather a replacement of a $d\theta/\theta$ distribution with a distribution that no longer has the $1/\theta$ divergence. Hadronisation of the heavy-quark will typically produce a heavy-quark hadron together with a couple of light hadrons. This will not restore the $d\theta/\theta$ distribution. (On the other hand, the authors are correct in stating that the decay products can fill the dead cone.)
- 4) In section 4, the authors state that they use "A selection of $k_T > \Lambda_{\text{QCD}}$ ($\Lambda_{\text{QCD}} = 200 \text{ MeV}/c$)". The quantity Λ_{QCD} is widely used in the literature to denote the precise scale that determines the magnitude of the strong coupling constant. As a solution, the authors could simply replace Λ_{QCD} with Λ , which would avoid the potential confusion.

5) In section 5, the manuscript refers to "the inverse logarithm of the angle". This should read "the logarithm of the inverse of the angle".

6) At the end of section 5, the manuscript mentions the ordering of the SHERPA and PYTHIA parton showers as being different. However, both showers use transverse momentum ordering. It would be more accurate to state that they "use different shower prescriptions and hadronisation models". In line with common practice in particle physics, the manuscript should also state the exact versions of all tools used (e.g. Pythia 8.303) and the details of the tune used for the generator parameters (e.g. the Monash 13 tune), as these can sometimes substantially affect the results.

7) The conclusions state "Experimental access to the property of quark masses has not been realised up until now, [...]. The only exception is the top quark [...]". I have two concerns with this statement:

a) any measurement of charm or bottom meson and baryon masses (of which there are very many) is, in my view, clear direct evidence of the charm and bottom masses, since those mesons masses are significantly larger than their light-quark counterparts. Indeed, such measurements provide the most precise determinations of heavy-quark masses, using lattice QCD to relate the meson masses to the quark masses. The authors could perhaps instead refer to "Experimental sensitivity to the masses of quasi-free quarks, prior to their binding into hadrons".

b) The manuscript mentions the exception of the top quark. They should also mention the bottom quark. There were several studies of the 3-jet fraction in b-tagged versus light-flavour jets at LEP and SLD, with resulting determinations of the b-quark mass. A list of references can be found in Table 1 of <https://arxiv.org/abs/2104.09924>. (Those extractions might be further improved by reanalysing the LEP and SLD data with a dead cone analysis of the kind employed in this manuscript.)

Referee #2 (Remarks to the Author):

A + B) Key results, originality, and significance.

The dead-cone effect is a fundamental and firm prediction of gauge field theories with massive charged particles. Several past experimental measurements have been consistent with its presence in QCD and inconsistent with its absence, so that, currently, I would not say there is any significant doubt regarding whether it exists or not. Nevertheless, a direct experimental measurement has been missing. This measurement is the first to claim to observe the effect directly, using fully reconstructed charm mesons whose decay products can be eliminated from the dead-cone region. This is a crucial new aspect with respect to previous studies. As such, I find this an important ground-breaking contribution to the field which to my mind deserves recognition. The measurement furthermore delivers new relevant constraints on theoretical attempts to model the effect, which will be of broad relevance to future studies involving heavy colour-charged emitters. I would recommend it for publication in Nature, but encourage the authors to address the suggestions and questions below.

C+D+E) Data & Methodology, statistics, uncertainties, and robustness.

The study exploits an established ([17]) technique for identifying ("tagging") jets containing D_0 charm mesons. A decay channel where the D_0 decays to two charged particles is used. These two charged particles are then artificially removed from the event and replaced by the D_0 , effectively "undoing" the D_0 decay. That alleviates one problem with earlier attempts to measure the dead-cone effect, where the decay products of the heavy hadron would effectively fill the dead-cone region, obscuring the effect. I find this to be a solid approach.

An iterative sequential-recombination jet-clustering algorithm well suited to this type of study (the Cambridge/Aachen algorithm) is then used on the constituents of the jet, to obtain an approximate picture of the leading sequence of perturbative emissions contributing to the substructure of the jet. In particular, this provides a classification of emissions originating from the charm quark, not without ambiguity but with at least the correct leading behaviour. One question I have here is whether use of an alternative algorithm with the same leading behaviour (like the k_T algorithm) would materially change the results? And why was the C/A algorithm preferred over, say, the k_T option? This may be due to the C/A one being more stable against noise from the underlying event, but the authors do not say so.

A nice complementarity is used to obtain a reference sample which, when charm is present, behaves in almost exactly the same way as the charm jets, by replacing the charm-tagged prong with the leading prong in the jet declustering. Here, I have a question. The reference sample will have some in principle unknown composition of leading quarks and leading

gluons. As touched on in the discussion of the reference variable $R(\theta)$ -no-dead-cone-limit, quark and gluon jets are different, eg since gluons have a roughly twice as large colour factor and hence emit twice as much radiation, and also since gluons have different collinear splitting kernels than quarks do. They are of course still massless and will not have a dead cone, so in that sense any linear combination of the two will presumably do fine as a reference sample for this measurement, but I did not find an explicit demonstration in the paper whether its conclusions have been shown to be stable against varying the reference assumption between, say, 100% quark jets, and 100% gluon jets. In fig 2, I think I can convince myself that even if I used pure light quarks or pure gluons, the difference (represented by the deviation of the dashed lines to unity) would not bridge the gap to the solid lines, but I think it would be good to make an explicit statement addressing this.

On p.6, to suppress hadronisation effects, a selection of $k_T > 200$ MeV is imposed. It is not really clear to me how this cut is imposed. Are all hadrons failing the cut discarded before running the jet-substructure clustering? If so, is the overall jet energy corrected for that loss? Or is something else done? Is k_T measured relative the jet axis, relative to the beam axis, or something else? Why does the statistical precision reduce if the cut is made higher? Does one lose some jets/events? Or does one just lose clusterings? Further, the authors do not seem to discriminate between hadronisation and underlying event effects here. The former would be present also in ee, the other would not. Both effects would indeed typically generate contributions of order Λ_{QCD} , but for hadronisation this is a correction, typically transverse to the jet axis (not the beam axis), whereas underlying-event effects would be more likely to be of absolute order Λ_{QCD} , with respect to the beam axis. Does the cut on $k_T > 200$ MeV approximately accomplish both? Or is one much more important than the other in this context?

Related to the above paragraph, I have a small pedantic correction to ask for. The authors write " $k_T > \Lambda_{\text{QCD}}$ ($\Lambda_{\text{QCD}} = 200$ MeV/c)" but that is not strictly correct (Λ_{QCD} is not exact equal to 200 MeV) and should be rephrased to $k_T > 200$ MeV $\sim \Lambda_{\text{QCD}}$ (or $= O(\Lambda_{\text{QCD}})$) or something equivalent, like low- k_T splittings at scales of order Λ_{QCD} , so that it is clear what exact cut is imposed (200 MeV) and that this is of order Λ_{QCD} if the authors are intent on highlighting that. (Same change in fig 2.) Additionally, I'm not sure whether it is helpful or confusing to mention that this corresponds to transverse distances smaller than the size of a hadron; that statement does not seem to me to add anything in the context since everything involved is in momentum space, but since it is not formally incorrect, I leave it to the authors' judgement whether to keep that clause or change it.

Also on p.6, the reference variable $R(\theta)$ -no-dead-cone-limit is defined. I find the subsequent discussion somewhat hard to follow. As mentioned, gluons radiate *more* than quarks, and their jets contain *more* hadrons, than quark jets do. Therefore, all else being equal, I would expect gluon jets to exhibit a higher $dn/d\ln(\theta)$ than quark jets, so my naive expectation would have been for $R < 1$. My guess is that since the jets are required to have comparable energies, the fact that the gluon jet energy is divided onto more particles mean that more of them fail the above mentioned k_T cut, ie although they contain more hadrons and hence more splittings in total, fewer of those remain after the cut? There is also a non-trivial normalisation mentioned "the θ distributions were normalised to the number of jets which contribute with n splittings". Is N_{DO} jets, for instance, equal to $n(1) + 2 n(2) + 3 n(3) + \dots$ where the number in parenthesis is the number of splittings (in that case it should not be given the superscript jets but rather splittings), or is it equal to $n(1) + n(2) + n(3) + \dots$? I believe the latter is the case, but it would be good to adopt a phrasing that eliminates any possible ambiguity, and here I found the "which contribute with n splittings" confusing, but would rather propose something like "the number of jets which contain at least one splitting".

I did not find a very precise definition of E_{radiator} , except in the caption of Fig 1. Given that this variable is used as a main classifier in the main result, fig 2, it would be useful to define it more precisely and comment on it, in the main text body, unless I missed it. It might also be helpful to make clear to the reader that, as I understand it, a single charm quark jet can actually populate all of the panes in fig 2, as the radiating charm quark loses progressively more of its energy during the (de)clustering. To be clear, I am not unhappy about this way of binning the results at all, I just think it could be good to make it a bit more crystal clear to the reader, eg, that E_{radiator} is not defined as a constant for each jet, say, but is defined for each individual clustering step.

I have no questions or comments on the statistical treatment. Few details are given, but I have no reason to doubt that this has followed professional good practice for a simple cut-based frequentist analysis, as standard within the ALICE collaboration and subject to strict internal reviews prior to publication.

G) References.

The main workhorse for corrections used in the analysis is the Pythia 6 generator. In addition to that program's overall manual which is cited, I think it would be correct in this study to cite the original physics paper in which its treatment of the dead-cone effect was derived and implemented, hep-ph/0010012. (The same treatment applies in Pythia 8, so kills two birds with one stone.)

F+H) Suggested improvements, clarity and context.

The second line in the abstract, "the vacuum is not transparent to the partons and induces ... a process that can be described as a parton shower", suggests a misconception about the nature of bremsstrahlung and should be corrected. When subjected to accelerations, even classical charges emit bremsstrahlung radiation. I find it misleading (or at least needlessly convoluted) to suggest that this has to do with whether the vacuum is transparent or not; that leads me to think about how partons interact with the nontrivial vacuum of quantum field theory. But bremsstrahlung including the dead cone is already there at the classical level.

An analogous rephrasing should be made in the second sentence in the introduction.

There are some other phrasings in the introduction which a pedant could also object to. These do not impact on the overall quality of the experimental measurement or its conclusions, but given this is submitted to Nature, I think these statements could also be made more clear and correct.

1) It is only in the strictly collinear limit that one can talk unambiguously about evolution of individual partons such as is done in the first paragraph. Outside that limit one has both coherence and recoil effects that intrinsically involve one or more other partons. But the paragraph does not make it clear that its reasoning is restricted to the collinear limit (or really, quasi-collinear limit, for massive particles). For the case studied in this paper, there is nothing wrong with considering that limit, it is just that this restriction is not currently mentioned in the first paragraph.

2) As above, the dead-cone effect is not only a fundamental feature of quantum gauge field theories, the $(1 - m^2/E^2/\theta^2)$ factor is there in classical gauge theories too (see eg Peskin & Schroeder ch 6.1). That could be made more clear.

3) The statement that the dead cone is due to the reduction in phase space for emission from a massive parton is confusing. Although the phase space does indeed become smaller, the dead cone can also be seen in the squared amplitude, in the form of negative mass terms. Thus, not only is the massive phase space somewhat smaller, there is also a reduced squared amplitude on it, which is near zero for angles less than the dead-cone one. One can perhaps perceive of the region with reduced amplitude as amounting to a "reduced phase space" but that would be unclear in my opinion, and somewhat misleading.

Referee #3 (Remarks to the Author):

This paper reports a measurement of the suppression of nearly collinear radiation from heavy flavor jets relative to light flavor jets, which is interpreted as an observation of the 'dead cone effect' generically predicated by quantum field theories with local gauge symmetries. To my knowledge, this is the first claim of a direct observation of the dead cone effect for the strong force. The paper is well-written and the result is an important contribution to high energy physics research. The declustering method is innovative. I would be happy to recommend publication following suitable answers to the following comments, questions, and suggestions.

1. Detector effects.

- Are detector effects used in the Pythia 6 simulation? I did not find any mention of Geant so I am unsure.

- Please provide the parameters used in Pythia 6, Pythia 8, Sherpa, etc. Also, please provide the details of how you turned off the dead cone effect. This is a generic feature of quantum field theory, so you cannot simply turn it off and have a sensible prediction. So there must be some approximation made and it would be good to give the specific parameters of the simulations used and briefly what approximation is being taken.

- Are the Pythia 8 and Sherpa samples passed through a detector simulation? If not, should I take Fig. 2 as corrected for detector effects (unfolded)? If that is the case, did I miss what unfolding method you are using? How do you define your truth target and fiducial volume?

2. Interpretation:

- What is the typical number of "emissions" per jet? How do you account for correlations between emissions from a single jet?

- "A lower limit for the significance of the small-angle suppression is estimated by comparing the measured data to $R = 1$..."
-> why is this a definitive test of the dead cone effect? First of all, isn't there a big contribution that is due to the "leading particle effect"? i.e. the fragmentation of charm quarks and light quarks and gluons is different. Also, couldn't this be due to other hadronization effects? You have found a significant difference in the ratio, but it could be due to many sources? I think the reason you think it is the dead cone is because of the simulations, right? This is not necessarily a problem, but it may be useful to modify the discussion and statistical analysis accordingly.

- Along the same lines, does the dead cone simulation on and off cause a big effect in other observables? There have been many measurements of jet shapes in heavy flavor and it would be useful to know if this would have already been seen in other standard measurements? I don't want to insist on a study that would be a lot of work, but given that you are making a strong claim in this paper, maybe it would be useful to use public Rivet routines to check with the Pythia and/or Sherpa simulations if this effect would have been seen elsewhere?

3. Wording:

- "...for charm and beauty quarks, which have masses of..." - seems strange to give such precise numbers with no error bars when the values themselves are presumably given in a scheme that is scale dependent (without the scale being precisely given).

- "...which has a mass of $1.86/c^2$..." -> citation

- It may be useful to cite other papers describing approaches for probing the dead cone effect in QCD.

- Ref. 14 is not a measurement (which is reserved for studies that correct for detector effects and include a full set of uncertainties) - please use a different word (perhaps "study"?)

Author Rebuttals to Initial Comments:

We thank the referees for their positive reviews and the insightful comments provided. Please find our answers inline.

Direct responses to questions are in red and any text modifications are presented in blue

Referees' comments:

Referee #1 (Remarks to the Author):

Report on "Direct observation of the dead-cone effect in QCD" by the ALICE collaboration.

This manuscript reports the direct observation of an effect in Quantum Chromodynamics (QCD) that was predicted long ago, but had yet to be decisively observed. That effect concerns the bremsstrahlung radiation from a fast-moving quark, and its partial suppression for massive quarks, specifically within angles (relative to the quark) of the order of the ratio of the quark rest mass to the quark energy.

The experimental analysis is clearly explained, and the final result in figure 2 is convincing, especially for lower quark ("radiator") energies. The study is backed up by an estimate of the statistical significance of the effect, which comfortably passes the threshold of 5 standard deviations that is commonly used in particle physics to claim "observation" of a new particle or phenomenon.

Given the importance of the phenomenon, a cornerstone of QCD, and the high quality of the analysis, I believe that the result meets the very high threshold required for publication in Nature. Before publication, however, some points need to be addressed / corrected in the manuscript's presentation of the general context, as outlined below.

Points to address:

1) Both the abstract and introduction make the statement that the "vacuum is not transparent to the partons and induces gluon radiation [...]". This is incorrect: it would be like saying that the vacuum is not transparent to electrons and that this induces photon radiation, which is patently false (non-transparency would imply that a free electron travelling through the vacuum would progressively lose energy and eventually come to a stop, which does not happen). Rather, the gluon radiation is the analogue of Bremsstrahlung in QED and is

induced by the change in direction (acceleration) of the parton when it scatters off another parton. These statements need to be corrected before the manuscript can be published.

We agree, the phrase used was incorrect. We have modified it consistently in the abstract and in Sec.1.

Abstract: These partons subsequently emit further partons in a process that can be described as a parton shower [2] which culminates in the formation of detectable hadrons.

Sec.1: In particle colliders, quarks and gluons are produced in high-energy interactions via processes with large momentum transfer, which are calculable and well described by QCD. These partons undergo subsequent emissions, resulting in the production of more quarks and gluons.

2) The statement, in the introduction that "the corresponding fraction carried by the leading hadron in light quark-initiated jets peaks at values close to zero" is correct only for case of extremely-energetic quark-initiated jets. For the energies being studied by the ALICE collaboration, the peak momentum fraction carried by the leading hadron is probably around 0.25 (the authors may want to verify this with a simulation, or simply drop the corresponding part of the sentence).

We checked some previous results from ALICE on light flavour FF and the distributions seem to be very peaked at values close to zero, even for low-pt jets. Below you can find a plot from <https://arxiv.org/pdf/1809.03232.pdf>. We nevertheless have changed the phrasing in the manuscript to "smaller values", which we find to be sufficiently general.

3) Still in the introduction, the statement that "the dead-cone angular region can be filled by hadronisation effects" is, I think incorrect. The dead cone is not a total absence of radiation, but rather a replacement of a $d\theta/\theta$ distribution with a distribution that no longer has the $1/\theta$ divergence. Hadronisation of the heavy-quark will typically produce a heavy-quark hadron together with a couple of light hadrons. This will not restore the $d\theta/\theta$ distribution. (On the other hand, the authors are correct in stating that the decay products can fill the dead cone.)

By "filled" we mean that the dead cone region is contaminated by this source, but certainly not that the divergent behaviour is recovered. We replaced "filled" by "receives contributions from" to remove the ambiguity.

4) In section 4, the authors state that they use "A selection of $k_T > \Lambda_{\text{QCD}}$ ($\Lambda_{\text{QCD}} = 200 \text{ MeV}/c$)," . The quantity Λ_{QCD} is widely used in the literature to denote the precise scale that determines the magnitude of the strong coupling constant. As a solution, the authors could simply replace Λ_{QCD} with Λ , which would avoid the potential confusion.

We have removed Λ_{QCD} entirely and replaced it with $k_T > 200 \text{ MeV}/c$.

5) In section 5, the manuscript refers to "the inverse logarithm of the angle". This should read "the logarithm of the inverse of the Angle".

Yes indeed this was incorrectly phrased. We have adjusted the manuscript.

6) At the end of section 5, the manuscript mentions the ordering of the SHERPA and PYTHIA parton showers as being different. However, both showers use transverse momentum ordering. It would be more accurate to state that they "use different shower prescriptions and hadronisation models". In line with common practice in particle physics, the manuscript should also state the exact versions of all tools used (e.g. Pythia 8.303) and the details of the tune used for the generator parameters (e.g. the Monash 13 tune), as these can sometimes substantially affect the results.

We agree and have amended the sentence on the shower prescriptions accordingly. We have also added the explicit MC versions and tunes to Sec.6.

Sec.5: SHERPA and PYTHIA are two MC generators commonly utilised in high-energy particle physics and use different shower prescriptions and hadronisation models. Both models implement the dead-cone effect.

7) The conclusions state "Experimental access to the property of quark masses has not been realised up until now, [...]. The only exception is the top quark [...]" . I have two concerns with this statement:

a) any measurement of charm or bottom meson and baryon masses (of which there are very many) is, in my view, clear direct evidence of the charm and bottom masses, since those mesons masses are significantly larger than their light-quark counterparts. Indeed, such measurements provide the most precise determinations of heavy-quark masses, using lattice QCD to relate the meson masses to the quark masses. The authors could perhaps instead refer to

"Experimental sensitivity to the masses of quasi-free quarks, prior to their binding into hadrons".

b) The manuscript mentions the exception of the top quark. They should also mention the bottom quark. There were several studies of the 3-jet fraction in b-tagged versus light-flavour jets at LEP and SLD, with resulting determinations of the b-quark mass. A list of references can be found in Table 1 of <https://arxiv.org/abs/2104.09924>. (Those extractions might be further improved by reanalysing the LEP and SLD data with a dead cone analysis of the kind employed in this manuscript.)

- a) We agree with the statement and we have reformulated the paragraph in the Sec.7 of the manuscript according to the suggestion provided.

Sec.7: By accessing the kinematics of the showering charm quark, prior to hadronisation, and directly uncovering the QCD dead-cone effect, our measurement provides direct sensitivity to the mass of quasi-free charm quarks, prior to their binding into hadrons.

- b) The reason we consider the top quark to be an exception is that one can directly measure the invariant mass of its reconstructed decay products. In the case of the 3 or 4-jet event fractions at LEP, one infers the b quark mass by comparing a NLO calculation to ratios of b-tagged to inclusive 3-jet rates, which is less direct than the aforementioned method for the top quark.

Referee #2 (Remarks to the Author):

A + B) Key results, originality, and significance.

The dead-cone effect is a fundamental and firm prediction of gauge field theories with massive charged particles. Several past experimental measurements have been consistent with its presence in QCD and inconsistent with its absence, so that, currently, I would not say there is any significant doubt regarding whether it exists or not. Nevertheless, a direct experimental

measurement has been missing. This measurement is the first to claim to observe the effect directly, using fully reconstructed charm mesons whose decay products can be eliminated from the dead-cone region. This is a crucial new aspect with respect to previous studies. As such, I find this an important ground-breaking contribution to the field which to my mind deserves recognition. The measurement furthermore delivers new relevant constraints on theoretical attempts to model the effect, which will be of broad relevance to future studies involving heavy colour-charged emitters. I would recommend it for publication in Nature, but encourage the authors to address the suggestions and questions below.

C+D+E) Data & Methodology, statistics, uncertainties, and robustness.

The study exploits an established ([17]) technique for identifying ("tagging") jets containing D0 charm mesons. A decay channel where the D0 decays to two charged particles is used. These two charged particles are then artificially removed from the event and replaced by the D0, effectively "undoing" the D0 decay. That alleviates one problem with earlier attempts to measure the dead-cone effect, where the decay products of the heavy hadron would effectively fill the dead-cone region, obscuring the effect. I find this to be a solid approach.

An iterative sequential-recombination jet-clustering algorithm well suited to this type of study (the Cambridge/Aachen algorithm) is then used on the constituents of the jet, to obtain an approximate picture of the leading sequence of perturbative emissions contributing to the substructure of the jet. In particular, this provides a classification of emissions originating from the charm quark, not without ambiguity but with at least the correct leading behaviour. One question I have here is whether use of an alternative algorithm with the same leading behaviour (like the kT algorithm) would materially change the results? And why was the C/A algorithm preferred over, say, the kT option? This may be due to the C/A one being more stable against noise from the underlying event, but the authors do not say so.

As described in Sec.2.4 of <https://arxiv.org/pdf/1807.04758.pdf> (and also in the first paragraph of Sec.3 in our manuscript) the choice of the C/A algorithm for the declustering is based on physical properties (the angular ordering, which is satisfied in nature) and analytical advantages concerning higher-order structures.

The k_T algorithm tends to cluster soft particles together first and this results in a different obtained jet tree, where for instance some emissions fail to appear in the primary Lund plane and are clustered in subleading prongs, as illustrated in Fig.3a of the reference provided. Therefore if the jet is reclustered with the k_T algorithm the splitting tree obtained will have a worse connection to the physics of the parton shower. For instance, the groomed momentum balance, z_g , obtained using k_T declustering is flat and does not exhibit the $1/z_g$ behaviour that is expected from its connection to the Altarelli Parisi splitting function. However this connection is recovered successfully when using the C/A algorithm, as shown in Fig.13 of <https://arxiv.org/pdf/1808.03689.pdf>.

A nice complementarity is used to obtain a reference sample which, when charm is present, behaves in almost exactly the same way as the charm jets, by replacing the charm-tagged prong with the leading prong in the jet declustering.

Here, I have a question. The reference sample will have some in principle unknown composition of leading quarks and leading gluons. As touched on in the discussion of the reference variable $R(\theta)$ -no-dead-cone-limit, quark and gluon jets are different, eg since gluons have a roughly twice as large colour factor and hence emit twice as much radiation, and also since gluons have different collinear splitting kernels than quarks do. They are of course still massless and will not have a dead cone, so in that sense any linear combination of the two will presumably do fine as a reference sample for this measurement, but I did not find an explicit demonstration in the paper whether its conclusions have been shown to be stable against varying the reference assumption between, say, 100% quark jets, and 100% gluon jets.

In fig 2, I think I can convince myself that even if I used pure light quarks or pure gluons, the difference (represented by the deviation of the dashed lines to unity) would not bridge the gap to the solid lines, but I think it would be good to make an explicit statement addressing this.

In Sec.6 of the manuscript we already mention explicitly that $R(\theta) = 1$ is the limit of $C_q = 1$. We also argue at the end of the Observable section that any realistic light quark fraction (i.e $C_q < 1$), leads to $R(\theta) > 1$. We have therefore not added an additional statement to the manuscript regarding this.

On p.6, to suppress hadronisation effects, a selection of $k_T > 200$ MeV is imposed. It is not really clear to me how this cut is imposed. Are all hadrons failing the cut discarded before running the jet-substructure clustering? If so, is the overall jet energy corrected for that loss? Or is something else done? Is k_T measured relative the jet axis, relative to the beam axis, or something else? Why does the statistical precision reduce if the cut is made higher? Does one loose some jets/events? Or does one just lose clusterings?

The cut on k_T is a cut on the k_T of each splitting. At each declustering step, we get two subjets, j_1, j_2 , characterized by an opening angle, θ , and a relative transverse momentum k_T (defined in the 2nd paragraph of Sec.3 in the manuscript). The leading subjet j_1 is a proxy for the heavy quark after the emission and the subleading prong j_2 is a proxy for the emitted gluon. In practice, $k_T = p_{T,j_2} \sin(\theta)$.

Therefore when we apply a k_T cut, we are removing splittings from the sample (these splittings are still reconstructed so as not to alter the splitting tree, but are just not accepted into the analysis). Naturally, low- k_T splittings dominate the sample, as the probability in QCD goes as $\alpha_s(k_T)$. That is why higher k_T cuts reduce the statistical reach of our measurement.

Further, the authors do not seem to discriminate between hadronisation and underlying event effects here. The former would be present also in ee , the other would not. Both effects would indeed typically generate contributions of order Λ_{QCD} , but for hadronisation this is a correction, typically transverse to the jet axis (not the beam axis), whereas underlying-event effects would be more likely to be of absolute order Λ_{QCD} , with respect to the beam axis. Does the cut on $k_T > 200$ MeV approximately accomplish both? Or is one much more important than the other in this context?

Both effects contribute to the Lund plane. Hadronisation effects contribute with extra splittings in the region of low- k_T , with no angular dependence. The impact in the Lund plane can be observed as a horizontal band at low k_T . The underlying event contributes by creating extra splittings, primarily at large angles. These few extra splittings at large-angles are the ones that are clustered last by the C/A algorithm, and as such do not interfere with the deepest structures of the tree that correspond to the smallest angles we are measuring. We have added a short remark in Sec.4 and removed Λ_{QCD} from the draft, replacing it simply with 200 MeV.

Sec.4: Other non-perturbative effects such as the underlying event contribute with extra soft splittings primarily at large angles and do not impact the small-angle region under study.

Related to the above paragraph, I have a small pedantic correction to ask for. The authors write " $k_T > \Lambda_{\text{QCD}}$ ($\Lambda_{\text{QCD}} = 200$ MeV/c)" but that is not strictly correct (Λ_{QCD} is not exact equal to 200 MeV) and should be rephrased to $k_T > 200$ MeV $\sim \Lambda_{\text{QCD}}$ (or = $O(\Lambda_{\text{QCD}})$) or something equivalent, like low- k_T splittings at scales of order Λ_{QCD} , so that it is clear what exact cut is imposed (200 MeV) and that this is of order Λ_{QCD} if the authors are intent on highlighting that. (Same change in fig 2.) Additionally, I'm not sure whether it is helpful or confusing to mention that this corresponds to transverse distances smaller than the size of a hadron; that statement does not seem to me to add anything in the context since everything involved is in momentum space, but since it is not formally incorrect, I leave it to the authors' judgement whether to keep that clause or change it.

We agree with the statement and have simplified the notation to $k_T > 200$ MeV. We added that this selects sufficiently hard splittings to suppress hadronisation effects. The comment on the inverse size of the hadron is indeed not necessary and as such we have removed it.

Also on p.6, the reference variable $R(\theta)$ no-dead-cone-limit is defined. I find the subsequent discussion somewhat hard to follow. As mentioned, gluons radiate more than quarks, and their jets contain more hadrons, than quark jets do. Therefore, all else being equal, I would expect gluon jets to exhibit a higher $dn/d\ln(\theta)$ than quark jets, so my naive expectation would have

been for $R < 1$. My guess is that since the jets are required to have comparable energies, the fact that the gluon jet energy is divided onto more particles mean that more of them fail the above mentioned k_T cut, ie although they contain more hadrons and hence more splittings in total, fewer of those remain after the cut? There is also a non-trivial normalisation mentioned "the theta distributions were normalised to the number of jets which contribute with n splittings". Is N_{D0} jets, for instance, equal to $n(1) + 2 n(2) + 3 n(3) + \dots$ where the number in parenthesis is the number of splittings (in that case it should not be given the superscript jets but rather splittings), or is it equal to $n(1) + n(2) + n(3) + \dots$? I believe the latter is the case, but it would be good to adopt a phrasing that eliminates any possible ambiguity, and here I found the "which contribute with n splittings" confusing, but would rather propose something like "the number of jets which contain at least one splitting".

Gluons indeed radiate more than quarks and one expects that the ratio of the total number of splittings per gluon vs quark jets is $9/4$ and this is roughly what one gets when the ratio is obtained with MC. However when we look at the (θ, k_T) distribution of splittings, there are differences in how the splittings are distributed with gluon emissions shifted to larger angles compared to quark emissions. The Sudakov form factor, which is the no radiation probability used in the MCs to dice the kinematics of the splitting, falls "faster" with the ordering scale for gluons than for quarks (see for instance Fig.2 of <https://arxiv.org/pdf/0710.3073.pdf>). Therefore gluons will emit more at the beginning of the evolution, populating the plane with large-angle emissions, while quarks will contribute with more splittings at small angles and high k_T . As gluons radiate more at large angles, when they reach the smaller angles, they have less available k_T . We have specified in the manuscript now that the $R(\theta) > 1$ behaviour occurs at small angles.

Concerning the normalisation, we indeed normalise by the number of jets that contribute with at least one splitting in the given E_{Radiator}, k_T bin, so we have modified the text according to the suggestion.

Sec.5: distributions were normalised to the number of jets which contain at least one splitting in the given E_{Radiator} and k_T selection, ...

Just to note, the number of times a jet contributes to a given E_{rad} bin is usually zero or one time. Therefore in practice, our normalisation is similar to normalising by the number of splittings.

I did not find a very precise definition of E_{radiator} , except in the caption of Fig 1. Given that this variable is used as a main classifier in the main result, fig 2, it would be useful to define it more precisely and comment on it, in the main text body, unless I missed it. It might also be helpful to make clear to the reader that, as I understand it, a single charm quark jet can actually populate all of the panes in fig 2, as the radiating charm quark loses progressively more of its energy during the (de)clustering. To be clear, I am not unhappy about this way of binning the results at all, I just think it could be good to make it a bit more crystal clear to the reader, eg, that E_{radiator} is not defined as a constant for each jet, say, but is defined for each individual clustering step.

We have explicitly added in Sec.3 of the manuscript that E_{radiator} is the sum of the energy of the two prongs at the given declustering step.

Sec.3: The angle between these splitting daughter prongs, θ , the relative transverse momentum of the splitting, k_T , and the sum of the energy of the two prongs, E_{radiator} , are registered.

I have no questions or comments on the statistical treatment. Few details are given, but I have no reason to doubt that this has followed professional good practice for a simple cut-based frequentist analysis, as standard within the ALICE collaboration and subject to strict internal reviews prior to publication.

:

G) References.

The main workhorse for corrections used in the analysis is the Pythia 6 generator. In addition to that program's overall manual which is cited, I think it would be correct in this study to cite the original physics paper in which its treatment of the dead-cone effect was derived and implemented, hep-ph/0010012. (The same treatment applies in Pythia 8, so kills two birds with one stone.)

We have added these to Sec.4 of the manuscript.

F+H) Suggested improvements, clarity and context.

The second line in the abstract, "the vacuum is not transparent to the partons and induces ... a process that can be described as a parton shower", suggests a misconception about the nature of bremsstrahlung and should be corrected. When subjected to accelerations, even classical charges emit bremsstrahlung radiation. I find it misleading (or at least needlessly convoluted) to suggest that this has to do with whether the vacuum is transparent or not; that leads me to think about how partons interact with the nontrivial vacuum of quantum field theory. But bremsstrahlung including the dead cone is already there at the classical level.

An analogous rephrasing should be made in the second sentence in the introduction.

We agree and have rephrased the sentence both in the abstract and in Sec.1.

Abstract: These partons subsequently emit further partons in a process that can be described as a parton shower [2] which culminates in the formation of detectable hadrons.

Sec.1: In particle colliders, quarks and gluons are produced in high-energy interactions via processes with large momentum transfer, which are calculable and well described by QCD. These partons undergo subsequent emissions, resulting in the production of more quarks and gluons.

There are some other phrasings in the introduction which a pedant could also object to. These do not impact on the overall quality of the experimental measurement or its conclusions, but given this is submitted to Nature, I think these statements could also be made more clear and correct.

1) It is only in the strictly collinear limit that one can talk unambiguously about evolution of individual partons such as is done in the first paragraph. Outside that limit one has both coherence and recoil effects that intrinsically involve one or more other partons. But the paragraph does not make it clear that its reasoning is restricted to the collinear limit (or really, quasi-collinear limit, for massive particles). For the case studied in this paper, there is nothing wrong with considering that limit, it is just that this restriction is not currently mentioned in the first paragraph.

We have modified the manuscript to specify that the parton cascade description is valid in the collinear limit.

Sec.1: This evolution can be described in the collinear limit by a cascade process known as a parton shower, that transfers the original parton energy to multiple lower energy particles.

2) As above, the dead-cone effect is not only a fundamental feature of quantum gauge field theories, the $(1 - m^2/E^2/\theta^2)$ factor is there in classical gauge theories too (see eg Peskin & Schroeder ch 6.1). That could be made more clear.

We have removed the word “quantum” from the expression “quantum field theory” in both the abstract and in Sec.1.

3) The statement that the dead cone is due to the reduction in phase space for emission from a massive parton is confusing. Although the phase space does indeed become smaller, the dead cone can also be seen in the squared amplitude, in the form of negative mass terms. Thus, not only is the massive phase space somewhat smaller, there is also a reduced squared amplitude on it, which is near zero for angles less than the dead-cone one. One can perhaps perceive of

the region with reduced amplitude as amounting to a "reduced phase space" but that would be unclear in my opinion, and somewhat misleading.

We have changed the phrase to "emission probability is suppressed".

Referee #3 (Remarks to the Author):

This paper reports a measurement of the suppression of nearly collinear radiation from heavy flavor jets relative to light flavor jets, which is interpreted as an observation of the 'dead cone effect' generically predicated by quantum field theories with local gauge symmetries. To my knowledge, this is the first claim of a direct observation of the dead cone effect for the strong force. The paper is well-written and the result is an important contribution to high energy physics research. The declustering method is innovative. I would be happy to recommend publication following suitable answers to the following comments, questions, and suggestions.

1. Detector effects.

- Are detector effects used in the Pythia 6 simulation? I did not find any mention of Geant so I am unsure.

Yes they were propagated through the ALICE setup using GEANT3. We have added a mention of this to Sec.4.

Sec.4: The studies were performed using Monte-Carlo (MC) PYTHIA 6.425 (Perugia 2011) [28,29] simulations (this generator includes mass effects in the parton shower [30] and was used for all MC-based corrections in this work), propagating the generated particles through a detailed description of the ALICE detector with GEANT3 [31].

- Please provide the parameters used in Pythia 6, Pythia 8, Sherpa, etc. Also, please provide the details of how you turned off the dead cone effect. This is a generic feature of quantum field theory, so you cannot simply turn it off and have a sensible prediction. So there must be some approximation made and it would be good to give the specific parameters of the simulations used and briefly what approximation is being taken.

We have updated the Sec.6 in the manuscript to highlight the specific versions and tunes of the MC.

Concerning turning the dead-cone effect on/off in the MC, we think there is a misunderstanding. Indeed there is not a simple way to turn off such an effect in the simulations. Instead our no dead-cone simulation corresponds simply to a light quark simulation, as shown in Eq.2, where sizeable mass effects are implicitly not present.

- Are the Pythia 8 and Sherpa samples passed through a detector simulation? If not, should I take Fig. 2 as corrected for detector effects (unfolded)? If that is the case, did I miss what unfolding method you are using? How do you define your truth target and fiducial volume?

We have clarified this further in the manuscript by adding a sentence to the beginning of Sec.6. Since our observable is a ratio, we verified that detector effects in the numerator and denominator cancel out to a large extent and we don't need to unfold. Besides, the available statistics wouldn't allow for a full 3D unfolding.

We verified that detector effects cancel out in the ratio via MC studies, where we compared the truth and detector-level $R(\theta)$ distributions and found them to be consistent. We verified this using a data-driven approach, described in the Methods section, where we considered the (dominant) impact of tracking efficiency on the data and rebuilt the ratio with an artificially reduced efficiency for both samples. Differences to the default ratio are considered in the systematic uncertainties.

Sec.6: Detector effects largely cancel out in the ratio and results are compared to particle-level simulations. Residual detector effects are considered in the systematic uncertainty together with uncertainties associated to...

2. Interpretation:

- What is the typical number of "emissions" per jet? How do you account for correlations between emissions from a single jet?

The average number of emissions per jet, at the low jet p_T considered, is about 5. We are unsure what exactly "correlations" is referring to in the second question, and thus have answered two points which we think could have been behind the question.

First we consider whether a couple of emissions can contribute to the same hadron. Hadronisation effects are subject to modelling. This is why we apply a high k_T cut, to suppress these effects and to make a more clear connection between what we call an emission and the gluons from the parton shower.

Second refers to the possibility that two gluon emissions can be packed together in the same prong. Such effects could happen equally in the heavy flavour and inclusive baseline samples and cancel out in the ratio. However, the C/A algorithm used for the reclustering avoids this problem as compared to for instance k_T reclustering, as explained in Fig.3 of <https://arxiv.org/pdf/1807.04758.pdf>.

- "A lower limit for the significance of the small-angle suppression is estimated by comparing the measured data to $R = 1$..." -> why is this a definitive test of the dead cone effect? First of all, isn't there a big contribution that is due to the "leading particle effect"? i.e. the fragmentation of charm quarks and light quarks and gluons is different. Also, couldn't this be due to other hadronization effects? You have found a significant difference in the ratio, but it could be due to many sources? I think the reason you think it is the dead cone is because of the simulations, right? This is not necessarily a problem, but it may be useful to modify the discussion and statistical analysis accordingly.

The quark and gluon fractions and their corresponding jet structures are not well described by the MC generators. There are also large differences in these descriptions between generators, in particular at low- p_T , as can be seen for instance in Fig.13 and Fig.14 in <https://cds.cern.ch/record/2759616/files/SMP-20-010-pas.pdf>.

For this reason we didn't want to compute a significance relative to a baseline that depends on the MC generators. Instead we considered the following: $R(\theta)=1$ is the limit for no dead-cone effect (where heavy-flavour sample behaves like a light-quark sample, as in Eq.2) if the inclusive jet sample consisted only of (light) quark jets (ie $C_q=1$).

Since our studies of the $R(\theta)$ distribution with q/inclusive show that the addition of gluons in the inclusive sample result in $R(\theta)>1$ at small angles (due to differences in the fragmentation of quarks and gluons), the limit of $R(\theta)=1$ represents a worst case scenario limit ($C_q=1$) that it is model independent.

Additionally the leading particle effect is intrinsically linked to the angle suppression we observe, and is thus part of the physics we are after.

- Along the same lines, does the dead cone simulation on and off cause a big effect in other observables? There have been many measurements of jet shapes in heavy flavor and it would be useful to know if this would have already been seen in other standard measurements? I don't want to insist on a study that would be a lot of work, but given that you are making a strong claim in this paper, maybe it would be useful to use public Rivet routines to check with the Pythia and/or Sherpa simulations if this effect would have been seen elsewhere?

As mentioned in the previous answer, the dead cone effect was not turned off in the MC as there is no clear way of doing so. Instead, we considered that the absence of mass effects in the $R(\theta)$ observable can be obtained by constructing this ratio with light quark over inclusive jet samples, as stated in Eq.2.

With this analysis we provide a measurement of the dead-cone effect directly by looking at $Q \rightarrow Qg$ splittings. However, of course this does not mean that the impact of the mass of the quark has not been measured before in other observables such as intrajet particle multiplicities or fragmentation functions, where the larger mass causes a reduction of the number of jet constituents and a hardening of the fragmentation as compared to light quarks.

3. Wording:

- "...for charm and beauty quarks, which have masses of.." - seems strange to give such precise numbers with no error bars when the values themselves are presumably given in a scheme that is scale dependent (without the scale being precisely given).

The uncertainties and the scheme used have been added to the manuscript in Sec.1.

- "...which has a mass of $1.86/c^2$..." -> citation

We have added the citation.

- It may be useful to cite other papers describing approaches for probing the dead cone effect in QCD.

We had indeed misplaced the reference to <https://arxiv.org/pdf/1606.03449.pdf> in the drafting process, which we have now added back.

- Ref. 14 is not a measurement (which is reserved for studies that correct for detector effects and include a full set of uncertainties) - please use a different word (perhaps "study"?)

We have reformulated the sentence and separated measurements (which we have now added additional references for) from the theoretical studies that link these measurements to the underlying QCD phenomena.

Sec.1: This is demonstrated by measurements such as the groomed momentum balance [14-17], that probes the DGLAP splitting function [19], and the Lund plane [20], which exposes the running of the strong coupling with the scale of the splittings.

Reviewer Reports on the First Revision:

Referees' comments:

Referee #1 (Remarks to the Author):

Report on revised version of "Direct observation of the dead-cone effect in QCD" by the ALICE collaboration.

The authors have, in my view, satisfactorily addressed the issues raised by all three referees and I can now recommend the article for publication in Nature. The only remaining comment that I have concerns the choice of references [1] (for QCD) and [2] (for parton showers):

[1] is a very old review (from 1978). If one is going to cite a review, then a more modern one might be of more help to the reader, for example that contained in the Particle Data Group's Review of Particle Physics (PTEP 2020 (2020) 8, 083C01).

[2] is a somewhat technical article. Keeping in mind the general readership of Nature, and the prominent position of this reference in the abstract, I wonder if <https://arxiv.org/abs/1101.2599> might be more appropriate, with its dedicated and pedagogical section 4 on parton showers

Referee #2 (Remarks to the Author):

I thank the authors for their comprehensive responses to the questions.

Referee #3 (Remarks to the Author):

Thank you for your detailed responses, they have cleared up my confusions. I believe the manuscript is nearly ready for publication, but I have a few of followup questions/points:

(1) Thank you for clarifying the impact of detector effects. Is the ALICE tracking efficiency independent of particle location (e.g. in eta), momentum, and the presence of nearby particles? If this is true, please state this in the text (or methods).

(2) Thank you for clarifying how the "no deadcone" was simulated; indeed, I misunderstood your baseline. However, I am still a little confused. Independent of the deadcone, aren't there differences between a charm jet and a light quark jet? How can I know that the differences you see are dominated by the dead cone effect and not something else? You make a strong argument that the angular location corresponding to the expectation based on the dead cone - can you maybe add a sentence or two that says that no other effects could have this scaling? (if true)

(3) Minor: B.2. - I still think that grouping [45] with a measurement ([19]) is misleading. There are a

variety of measurements from CMS and ATLAS that use tracks (not just [19]) with declustering and I would reference those as measurements and if you insist on citing [45], then I would refer to it as a study (you say study now, but then follow that with "measurement" a few words later).

Author Rebuttals to First Revision:

Last comments from the Nature Referees

We thank all three referees for all their comments and suggestions during the review process that helped improve the manuscript.

Referees' comments:

Referee #1 (Remarks to the Author):

Report on revised version of "Direct observation of the dead-cone effect in QCD" by the ALICE collaboration.

The authors have, in my view, satisfactorily addressed the issues raised by all three referees and I can now recommend the article for publication in Nature. The only remaining comment that I have concerns the choice of references [1] (for QCD) and [2] (for parton showers):

[1] is a very old review (from 1978). If one is going to cite a review, then a more modern one might be of more help to the reader, for example that contained in the Particle Data Group's Review of Particle Physics (PTEP 2020 (2020) 8, 083C01).

[2] is a somewhat technical article. Keeping in mind the general readership of Nature, and the prominent position of this reference in the abstract, I wonder if <https://arxiv.org/abs/1101.2599> might be more appropriate, with its dedicated and pedagogical section 4 on parton showers

We have added both reference suggestions to the manuscript.

Referee #2 (Remarks to the Author):

I thank the authors for their comprehensive responses to the questions.

Referee #3 (Remarks to the Author):

Thank you for your detailed responses, they have cleared up my confusions. I believe the manuscript is nearly ready for publication, but I have a few of followup questions/points:

(1) Thank you for clarifying the impact of detector effects. Is the ALICE tracking efficiency independent of particle location (e.g. in η), momentum, and the presence of nearby particles? If this is true, please state this in the text (or methods).

In pp collisions, the tracking efficiency in ALICE is constant at approximately 80% for particles with $p_T > 500$ MeV/c, and remains homogeneous as a function of pseudorapidity and azimuthal angle. Track density effects, at the levels encountered in pp collisions, are also negligible. We

have added ALICE ref.[45] that details the ALICE detector performance and shows negligible tracking efficiency differences as function of track density by comparing pp and PbPb collisions

(2) Thank you for clarifying how the "no deadzone" was simulated; indeed, I misunderstood your baseline. However, I am still a little confused. Independent of the deadcone, aren't there differences between a charm jet and a light quark jet? How can I know that the differences you see are dominated by the dead cone effect and not something else? You make a strong argument that the angular location corresponding to the expectation based on the dead cone - can you maybe add a sentence or two that says that no other effects could have this scaling? (if true)

The only element that dictates differences between light-quark and heavy-quark showers is the mass of the quark, as light and heavy quarks are otherwise identical for all properties regarding QCD showers. There are possible differences between the two sets arising at the hadronisation stage, however the k_T selections applied in the analysis remove such hadronisation effects. Also the hadronisation scale is different, but this we take into account by the leading particle requirement. We think that eq.2 and the text above stress this point sufficiently so we haven't added extra text.

(3) Minor: B.2. - I still think that grouping [45] with a measurement ([19]) is misleading. There are a variety of measurements from CMS and ATLAS that use tracks (not just [19]) with declustering and I would reference those as measurements and if you insist on citing [45], then I would refer to it as a study (you say study now, but then follow that with "measurement" a few words later). We have replaced citation [45] by a CMS measurement of jet substructure using declustering in $t\bar{t}$ events(<https://arxiv.org/abs/1808.07340>). In this way we now have two examples, one from ATLAS and one from CMS, for track-based observables.